# Adversarially Robust Dense-Sparse Tradeoffs via Heavy-Hitters

**David P. Woodruff**
Department of Computer Science
Carnegie Mellon University
dwoodruf@andrew.cmu.edu

**Samson Zhou**
Department of Computer Science
Texas A&M University
samsonzhou@gmail.com

## Abstract

In the adversarial streaming model, the input is a sequence of adaptive updates that defines an underlying dataset and the goal is to approximate, collect, or compute some statistic while using space sublinear in the size of the dataset. In 2022, Ben-Eliezer, Eden, and Onak showed a dense-sparse trade-off technique that elegantly combined sparse recovery with known techniques using differential privacy and sketch switching to achieve adversarially robust algorithms for $L_p$ estimation and other algorithms on turnstile streams. However, there has been no progress since, either in terms of achievability or impossibility. In this work, we first give improved algorithms for adversarially robust $L_p$-heavy hitters, utilizing deterministic turnstile heavy-hitter algorithms with better tradeoffs. We then utilize our heavy-hitter algorithm to reduce the problem to estimating the frequency moment of the tail vector. We give a new algorithm for this problem in the classical streaming setting, which achieves additive error and uses space independent in the size of the tail. We then leverage these ingredients to give an improved algorithm for adversarially robust $L_p$ estimation on turnstile streams. We believe that our results serve as an important conceptual message, demonstrating that there is no inherent barrier at the previous state-of-the-art.

## 1 Introduction

Adversarial robustness for big data models is increasingly important not only for ensuring the reliability and security of algorithmic design against malicious inputs and manipulations, but also to retain guarantees for honest inputs that are nonetheless co-dependent with previous outputs of the algorithm. One such big data model is the streaming model of computation, which has emerged as a central paradigm for studying statistics of datasets that are too large to store. Common examples of datasets that are well-represented by data streams include database logs generated from e-commerce transactions, Internet of Things sensors, scientific observations, social network traffic, or stock markets. To capture these applications, the one-pass streaming model defines an underlying dataset that evolves over time through a number of sequential updates that are discarded irrevocably after processing, and the goal is to compute or approximate some fixed function of the dataset while using space sublinear in both the length $m$ of the data stream and the dimension $n$ of the dataset.

**The streaming model of computation.** In the classical *oblivious* streaming model, the stream $S$ of updates $u_1, \ldots, u_m$ defines a dataset that is fixed in advance, though the ordering of the sequence of updates may be adversarial. In other words, the dataset is oblivious to any algorithmic design choices, such as instantiations of internal random variables. This is vital for many streaming algorithms, which crucially leverage randomness to achieve meaningful guarantees in sublinear space. For example, the celebrated AMS sketch [AMS99] initializes a random sign vector $s$ and outputs $|\langle s, f \rangle|$ as the estimate for the $L_2$ norm of the underlying frequency vector $f$ defined by the stream. To show

38th Conference on Neural Information Processing Systems (NeurIPS 2024).

correctness of the sketch, we require $s$ to be chosen uniformly at random, independent of the value of $f$. Similar assumptions are standard across many fundamental sublinear algorithms for machine learning, such as linear regression, low-rank approximation, or column subset selection.

Unfortunately, such an assumption is unreasonable [MNS11, GHS+12, BMSC17, NY19, CN20], as an honest user may need to repeatedly interact with an algorithm, choosing their future actions based on responses to previous questions. For example, in recommendation systems, it is advisable to produce suggestions so that when a user later decides to dismiss some of the items previously recommended by the algorithm, a new high-quality list of suggestions can be quickly computed without solving the entire problem from scratch [KMGG07, MBN+17, KZK18, OSU18, AMYZ19]. Another example is in stochastic gradient descent or linear programming, where each time step can update the eventual output by an amount based on a previous query. For tasks such as linear regression, actions as simple as sorting a dataset have been shown to cause popular machine learning libraries to fail [BHM+21].

**Adversarially robust streaming model.** In the adversarial streaming model [BY20, HKM+20, ABD+21, BHM+21, KMNS21, WZ21b, BKM+22, BEO22, BJWY22, CGS22, ACGS23, ACSS23, DSWZ23, GLW+24], a sequence of adaptively chosen updates $u_1, \ldots, u_m$ is given as an input data stream to an algorithm. The adversary may choose to generate future updates based on previous outputs of the algorithm, while the goal of the algorithm is to correctly approximate or compute a fixed function at all times in the stream. Formally, the *black-box* adversarial streaming model can be modeled as a two-player game between a streaming algorithm $\mathcal{A}$ and a source $\mathfrak{E}$ that creates a stream of adaptive and possibly adversarial inputs to $\mathcal{A}$. Prior to the game, a fixed statistic $\mathcal{Q}$ is determined, so that the goal of the algorithm is to approximate $\mathcal{Q}$ on the sequence of inputs seen at each time. The game then proceeds across $m$ rounds. In the $t$-th round:

(1) $\mathfrak{E}$ computes an update $u_t$ for the stream, which possibly depends on all previous outputs from $\mathcal{A}$.

(2) $\mathcal{A}$ uses $u_t$ to update its data structures $\mathcal{D}_t$, acquires a fresh batch $R_t$ of random bits, and outputs a response $Z_t$ to the query $\mathcal{Q}$.

(3) $\mathfrak{E}$ observes and records the response $Z_t$.

The goal of $\mathfrak{E}$ is to induce from $\mathcal{A}$ an incorrect response $Z_t$ to the query $\mathcal{Q}$ at some time $t \in [m]$ throughout the stream using its control over the sequence $u_1, \ldots, u_m$. By the nature of the game, only a single pass over the stream is permitted. In the context of our paper, each update $u_t$ has the form $(i_t, \Delta_t)$, where $i_t \in [n]$ and $\Delta_t \in \{\pm 1\}$. The updates implicitly define a frequency vector $f \in \mathbb{R}^n$, so that $u_t$ changes the value of the $(i_t)$-th coordinate of $f$ by $\Delta_t$.

**Turnstile streams and flip number.** In the turnstile model of streaming, updates are allowed to either increase or decrease the weight of elements in the underlying dataset, as compared to insertion-only streams, where updates are only allowed to increase the weight. Whereas various techniques are known for the adversarial robustness on insertion-only streams, significantly less is known for turnstile streams. While near-optimal adversarially robust streaming algorithms for fundamental problem such as $L_p$ estimation have been achieved in polylogarithmic space for $p \leq 2$ by [WZ21b] in the insertion-only model, it is a well-known open question whether there exists a constant $C = \Omega(1)$ such that the same problems require space $\Omega(n^C)$, where $n$ is the dimension of the underlying frequency vector. Indeed, [HW13] showed that the existence of a constant $C = \Omega(1)$ such that no linear sketch with sketching dimension $o(n^C)$ can approximate the $L_2$ norm of an underlying frequency vector within even a polynomial multiplicative factor, when the adversarial input stream is turnstile and real-valued.

Given an accuracy parameter $(1 + \varepsilon)$, the *flip number* $\lambda$ is the number of times the target function $\mathcal{Q}$ changes by a factor of $(1 + \mathcal{O}(\varepsilon))$. It is known that for polynomially-bounded monotone functions $\mathcal{Q}$ on insertion-only streams, we generally have $\lambda = \mathcal{O}\left(\frac{1}{\varepsilon} \log m\right)$, but for turnstile streams that toggle the underlying frequency vector between the all-zeros vector and a nonzero vector with each update, we may have $\lambda = \Omega(m)$. There are various techniques that then implement $\lambda$ [BJWY22] or even roughly $\sqrt{\lambda}$ [HKM+20, ACSS23] independent instances of an oblivious algorithm, processing all stream updates to all instances. Therefore, the space complexity of these approaches are at least roughly $\sqrt{\lambda}$ times the space required by the oblivious algorithm, which may not be desirable in large setting of $\lambda = \Omega(m)$ for turnstile streams. By considering *dense-sparse tradeoffs*, [BEO22] gave a general

framework that improved upon the $\tilde{\mathcal{O}}\left(\sqrt{\lambda}\right) = \tilde{\mathcal{O}}\left(\sqrt{m}\right)$ space bounds due to the flip number. In particular, their results show that $\tilde{\mathcal{O}}\left(m^{p/(2p+1)}\right)$ space suffices for the goal of $L_p$ norm estimation, where the objective is to estimate $(f_1^p + \ldots + f_n^p)^{1/p}$ for an input vector $f \in \mathbb{R}^n$, which is an important problem that has a number of applications, such as network traffic monitoring [FKSV02, KSZC03, TZ04], clustering and other high-dimensional geometry problems [BIRW16, CJLW22, CCJ$^+$23, CWZ23], low-rank approximation and linear regression [CW09, FMSW10, BDM$^+$20, VVWZ23, WY23], earth-mover estimation [Ind04, AIK08, ABIW09], cascaded norm estimation [JW09, MRWZ20], and entropy estimation [HNO08]. Unfortunately, there has been no progress for $L_p$ estimation on turnstile streams since the work of [BEO22], either in terms of achievability or impossibility. Thus we ask:

> Is there a fundamental barrier for adversarially robust $L_p$ estimation on turnstile streams beyond the dense-sparse tradeoffs?

## 1.1 Our Contributions

In this paper, we answer the above question in the negative. We show that the techniques of [BEO22] do not realize a fundamental limit for adversarially robust $L_p$ estimation on turnstile streams. In particular, we give an algorithm that uses space $\tilde{\mathcal{O}}\left(m^c\right)$, for some constant $c < \frac{p}{2p+1}$ for $p \in (1, 2)$. We first require an adversarially robust algorithm for heavy-hitters.

**Heavy hitters.** Recall that the $\varepsilon$-$L_p$-heavy hitter problem is defined as follows.

**Definition 1.1** ($\varepsilon$-$L_p$-heavy hitters). *Given a vector $f \in \mathbb{R}^n$ and a threshold parameter $\varepsilon \in (0, 1)$, output a list $\mathcal{L}$ that includes all $i$ such that $f_i \geq \varepsilon \cdot \|f\|_p$ and includes no $j$ such that $f_j < \frac{\varepsilon}{2} \cdot \|f\|_p$.*

Generally, heavy-hitter algorithms actually solve the harder problem of outputting a estimated frequency $\widehat{f}_i$ such that $|\widehat{f}_i - f_i| \leq C \cdot \varepsilon \cdot \|f\|_p$, for each $i \in [n]$, where $C$ is some constant such as $\frac{1}{6}$. Observe that such a guarantee solves the $\varepsilon$-$L_p$-heavy hitters problem because each $i$ such that $f_i \geq \varepsilon \cdot \|f\|_p$ must have $\widehat{f}_i > \frac{3\varepsilon}{4} \cdot \|f\|_p$ and similarly each $j$ such that $\widehat{f}_j \geq \frac{3\varepsilon}{4}$ must have $f_j \geq \frac{\varepsilon}{2} \cdot \|f\|_p$. We give an adversarially robust streaming algorithm for the $L_p$-heavy hitters problem on turnstile streams.

**Theorem 1.2.** *Let $p \in [1, 2]$. There exists an algorithm that uses $\tilde{\mathcal{O}}\left(\frac{1}{\varepsilon^{2.5}} m^{(2p-2)/(4p-3)}\right)$ bits of space and solves the $\varepsilon$-$L_p$-heavy hitters problem at all times in an adversarial stream of length $m$.*

Though not necessarily obvious, our result in Theorem 1.2 improves on the dense-sparse framework of [BEO22] across all $p \in [1, 2)$. Specifically, the result of [BEO22] uses space $\tilde{\mathcal{O}}\left(m^\alpha\right)$ for $\alpha = \frac{p}{2p+1}$, while our result uses space $\tilde{\mathcal{O}}\left(m^\beta\right)$ for $\beta = \frac{2p-2}{4p-3}$. It can be shown that $\alpha - \beta = \frac{2-p}{(4p-3)(2p+1)}$, which is at positive for all $p \in [1, 2)$. Thus our result is an important conceptual contribution showing that the true nature of the heavy-hitter problem lies beyond the techniques of [BEO22].

A particular regime of interest is $p = 1$, where the previous dense-sparse framework of [BEO22] achieves $\tilde{\mathcal{O}}\left(m^{1/3}\right)$ bits of space, but our result in Theorem 1.2 only requires polylogarithmic space.

**Moment estimation.** Along the way to our main result, we also give a new algorithm for estimating the residual of a frequency vector up to some tail error. More precisely, given a frequency vector $f$ that is defined implicitly through a data stream and a parameter $k > 0$, let $g$ be a tail vector of $f$, which omits the $k$ entries of $f$ largest in magnitude, breaking ties arbitrarily. Similarly, let $h$ be a tail vector of $f$ that omits the $(1 - \varepsilon)k$ entries of $f$ largest in magnitude, where $\varepsilon \in (0, 1)$ serves as an error parameter. Then we give a one-pass streaming algorithm that outputs an estimate for $\|g\|_p^p$ up to additive $\varepsilon \cdot \|h\|_p^p$, using space poly $\left(\frac{1}{\varepsilon}, \log n\right)$. In particular, our space is independent of the tail parameter $k$. We defer the full guarantees of our algorithm as well as a more formal discussion to Section 3. We then give our main result:

**Theorem 1.3.** *Let $p \in [1, 2]$ and $c = \frac{24p^2 - 23p + 4}{(4p-3)(12p+3)}$. There exists a streaming algorithm that uses $\mathcal{O}\left(m^c\right) \cdot \text{poly}\left(\frac{1}{\varepsilon}, \log(nm)\right)$ bits of space and outputs a $(1 + \varepsilon)$-approximation to the $L_p$ norm of the underlying vector at all times of an adversarial stream of length $m$.*

It can again be shown that our result in Theorem 1.3 again improves on the dense-sparse framework of [BEO22] across all $p \in (1, 2)$. For example, for $p = 1.5$, the previous result uses space $\tilde{\mathcal{O}}\left(m^{3/8}\right) = \tilde{\mathcal{O}}\left(m^{0.375}\right)$, while our algorithm uses space $\tilde{\mathcal{O}}\left(m^{47/126}\right) \approx \tilde{\mathcal{O}}\left(m^{0.373}\right)$. Although our quantitative improvement is mild, we believe it nevertheless illustrates an important message which shows that the dense-sparse technique does not serve as an impossibility barrier.

## 1.2 Technical Overview

Recall that the *flip number* $\lambda$ to be the number of times the $F_p$ moment changes by a factor of $(1 + \mathcal{O}(\varepsilon))$, given a target accuracy $(1 + \varepsilon)$. Given a stream with flip number $\lambda$, the standard *sketch-switching* technique [BJWY22] for adversarial robustness is to implement $\lambda$ independent instances of an oblivious streaming algorithm for $F_p$ estimation, iteratively using the output of each algorithm only when it differs from the output of the previous algorithm by a $(1 + \varepsilon)$-multiplicative factor. Subsequently, [HKM+20, ACSS23] showed that by using differential privacy, it suffices to use roughly $\sqrt{\lambda}$ independent instances of an oblivious streaming algorithm for $F_p$ estimation to achieve correctness at all times for an adaptive input stream. Unfortunately, the flip number for a stream of length $m$ can be as large as $\Omega(m)$, such as in the case where the underlying frequency vector alternates between the all zeros vector and a nonzero vector.

The dense-sparse framework of [BEO22] observes that the only case where the flip number can be large is when there are a large number of times in the stream where the corresponding frequency vector is somewhat sparse. For example, in the above scenario where the underlying frequency vector alternates between the all zeros vector and a nonzero vector, all input vectors are 1-sparse. In fact, they notice that for $F_p$ estimation, that once the frequency vector has at least $m^C$ nonzero entries for any fixed constant $C \in (0, 1)$, then since all entries must be integral and all updates only change each entry by 1, at least $\Omega_\varepsilon(m^{C/p})$ updates are necessary before the $p$-th moment of the resulting frequency vector can differ by at least $(1 + \varepsilon)$-multiplicative factor. Hence in the stream updates where the frequency vector has at least $m^C$ nonzero entries, the flip number can be at most $\mathcal{O}\left(m^{1-C/p}\right)$, for $\varepsilon = \Omega(1)$. Thus it suffices to run $\tilde{\mathcal{O}}\left(m^{1/2-C/2p}\right)$ independent instances of the oblivious algorithm, using the differential privacy technique of [HKM+20, ACSS23]. Moreover, in the case where the vector is $m^C$-sparse, there are sparse recovery techniques that can exactly recover all the nonzero coordinates using $\tilde{\mathcal{O}}\left(m^C\right)$ space, even if the input is adaptive. Hence by balancing $\tilde{\mathcal{O}}\left(m^C\right) = \tilde{\mathcal{O}}\left(m^{(1/2-C/2p)}\right)$ at $C = \frac{1}{3}$, [BEO22] achieves $\tilde{\mathcal{O}}\left(m^{p/(2p+1)}\right)$ overall space for $F_p$ estimation for adaptive turnstile streams.

Our key observation is that for $p \in (1, 2)$, if the frequency vector has at least $m^C$ nonzero entries, a sequence of $\mathcal{O}_\varepsilon(m^{C/p})$ updates may not always change the $p$-th moment of the underlying vector. For example, if the updates are all to separate coordinates, then the $p$-th moment may actually change very little. In fact, a sequence of $\mathcal{O}_\varepsilon(m^{C/p})$ updates may *only* change the $p$-th moment of the underlying vector by $(1 + \varepsilon)$ if most of the updates are to a small number of coordinates. As a result, most of the updates are to some coordinate that was either initially a heavy-hitter or subsequently a heavy-hitter. Then by tracking the heavy-hitters of the underlying frequency vector, we can handle the hard input for [BEO22], thus demanding a larger number of stream updates before the $p$-th moment of the vector can change by $(1 + \varepsilon)$. Consequently, the number of independent instances decreases, which facilitates a better balancing and allows us to achieve better space bounds. Unfortunately, there are multiple challenges to realizing this intuition.

**Heavy-hitters.** First, we need a streaming algorithm for accurately reporting the frequencies of the $L_p$-heavy hitters at all times in the adaptive stream. However, such a subroutine is not known and naïvely, one might expect an estimate of the $L_p$ norm might be necessary to identify the $L_p$ heavy-hitters. Moreover, algorithms for finding $L_p$ heavy-hitters are often used to estimate the $L_p$ norm of the underlying frequency, e.g., [IW05, WZ12, BBC+17, LSW18, BWZ21, WZ21a, MWZ22, BMWZ23, JWZ24]. Instead, we use a turnstile streaming algorithm DETHH for $L_p$ heavy-hitters [GM07] that uses sub-optimal space $\tilde{\mathcal{O}}\left(\frac{1}{\varepsilon^2} \cdot n^{2-2/p}\right)$ bits of space for $p \in (1, 2]$, rather than the optimal COUNTSKETCH, which uses $\mathcal{O}\left(\frac{1}{\varepsilon^2} \cdot \log^2 n\right)$ bits of space. However, the advantage of DETHH is that the algorithm is deterministic, so we can utilize the previous intuition from the dense-sparse framework of [BEO22]. In particular, if the universe size is small, then we can run DETHH, and if the universe size is large, then we collectively handle these cases using an ensemble

of COUNTSKETCH algorithms via differential privacy. We provide the full details of the robust $L_p$-heavy hitter algorithm in Section 2, ultimately achieving Theorem 1.2.

**Residual estimation.** It remains to estimate the contribution of the elements that are not $L_p$ heavy-hitters, i.e., the residual vector, toward the overall $p$-th moment. More generally, given a tail parameter $k > 0$ and an error parameter $\varepsilon \in (0, 1)$, let $g$ be a tail vector of $f$ that omits the $k$ entries of $f$ largest in magnitude, breaking ties arbitrarily and let $h$ be a tail vector of $f$ that omits the $(1 - \varepsilon)k$ entries of $f$ largest in magnitude. We define the level sets of the $p$-th moment so that level set $\Lambda_\ell$ roughly consists of the coordinates of $g$ with magnitude $[(1 + \varepsilon)^\ell, (1 + \varepsilon)^{\ell+1})$. We then estimate the contribution of each level set to the $p$-th moment of the residual vector using the subsampling framework introduced by [IW05].

Namely, we note that any "significant" level set has either a small number of items with large magnitude, or a large number of items that collectively have significant contribution to the $p$-th moment. In the former case, we can use COUNTSKETCH to identify the items with large magnitude, while in the latter case, it can be shown that after subsampling the universe, there will be a large number of items in the level set that remain. Moreover, these items will now be heavy with respect to the $p$-th moment of the resulting frequency vector after subsampling with high probability. Thus, these items can be identified by COUNTSKETCH on the subsampled universe. Furthermore, after rescaling inversely by the sampling probability, the total number of such items in the level set can be estimated accurately by rescaling the number of the heavy-hitters in the subsampled universe. Hence in both cases, we can estimate the number of items in the significant level sets and subtract off the largest $k$ such items. We provide the full details of the residual estimation algorithm in Section 3, culminating in Theorem 3.4.

## 2  Adversarially Robust $L_p$-Heavy Hitters

In this section, we give an adversarially robust algorithm for $L_p$-heavy hitters on turnstile streams. We first recall the following deterministic algorithm for $L_p$-heavy hitters on turnstile streams.

**Theorem 2.1.** *[GM07] For $p \in [1, 2)$, there exists a deterministic algorithm* DETHH *that solves the $\varepsilon$-$L_p$ heavy-hitters on a universe of size $t$ and a stream of length $m$ and uses $\frac{1}{\varepsilon^2} t^{2-2/p}$ polylog $\frac{tm}{\varepsilon}$ bits of space.*

We also recall the following variant of COUNTSKETCH for answering a number of rounds of adaptive queries, as well as a more general framework for answering adaptive queries.

**Theorem 2.2.** *[CLN+22] For $p \in [1, 2)$, there exists a randomized algorithm* ROBUSTCS *that uses $\tilde{\mathcal{O}}\left(\frac{\sqrt{\lambda}}{\varepsilon^2} \log n \log \frac{nm\lambda}{\delta}\right)$ bits of space, and for $\lambda$ different times $t$ on an adaptive stream of length $m$ on a universe of size $n$, reports for all $i \in [n]$ an estimate $\widehat{f_i^{(t)}}$ such that $|\widehat{f_i^{(t)}} - f_i^{(t)}| \le \frac{\varepsilon}{100} \cdot \|f^{(t)}\|_2$, where $f^{(t)}$ is the induced frequency vector at time $t$.*

**Theorem 2.3.** *[HKM+20, BKM+22, ACSS23, CSW+23] Given a streaming algorithm $\mathcal{A}$ that uses $S$ space and answers a query with constant failure probability $\delta_0 < \frac{1}{2}$, there exists a data structure that answers $Q$ adaptive queries, with probability $1 - \delta$ using space $\mathcal{O}\left(S\sqrt{Q} \log^2 \frac{Q}{\delta}\right)$.*

While ROBUSTCS has better space guarantees than DETHH, determinism nevertheless serves an important purpose for us. Namely, adversarial input can induce failures on randomized algorithms but cannot induce failures on deterministic algorithms. On the other hand, the space usage of DETHH grows with the size of the universe. Thus, we now use insight from the dense-sparse framework of [BEO22]. If the universe size is small, then we shall use DETHH. On the other hand, if the universe size is large, then shall use the following robust version of COUNTSKETCH, requiring roughly $\sqrt{\lambda}$ number of independent instances, where $\lambda$ is the flip number. The key observation is that because the universe size is large, then the flip number will be much smaller than in the worst possible case. Moreover, we can determine which case we are in, i.e., the large universe case or the small universe case, by using the following $L_0$ estimation algorithm:

**Theorem 2.4.** *[KNW10] There exists an insertion-deletion streaming algorithm* LZEROEST *that uses $\mathcal{O}\left(\frac{1}{\varepsilon^2} \log n \log \frac{1}{\delta} \left(\log \frac{1}{\varepsilon} + \log \log m\right)\right)$ bits of space, and with probability at least $1 - \delta$, outputs a $(1 + \varepsilon)$-approximation to $L_0$.*

---

**Algorithm 1** ROBUSTHH: Adversarially robust $L_p$-heavy hitters

---

**Input:** Turnstile stream of length $m$ for a frequency vector of length $n$
**Output:** Adversarially robust heavy-hitters

1: $t \leftarrow \mathcal{O}\left(m^{p/(4p-3)}\right), \ell \leftarrow \frac{\varepsilon}{100} \cdot t^{1/p}, b \leftarrow \frac{m}{\ell}$, STATE $\leftarrow$ SPARSE
2: Initialize DETHH with threshold $\frac{\varepsilon}{16}$
3: Initialize ROBUSTCS robust to $b$ queries, with threshold $\frac{\varepsilon}{16}$ for $r = \mathcal{O}\left(\frac{m}{\varepsilon t^{1/p}}\right)$ rounds
4: Initialize $\tilde{\mathcal{O}}\left(\sqrt{b}\right)$ copies LZEROEST with accuracy 2 robust to $b$ queries
5: **for** each block of $\ell$ updates **do**
6:     Update DETHH, ROBUSTCS, and all copies of LZEROEST
7:     **if** STATE = SPARSE at the beginning of the block **then**
8:         Return the output of DETHH
9:     **else**
10:         Return the output of ROBUSTCS at the beginning of the block     ▷Theorem 2.2
11:     Let $Z$ be the output of robust LZEROEST     ▷Theorem 2.3 and Theorem 2.4
12:     **if** $Z > 100t$ **then**
13:         STATE $\leftarrow$ DENSE
14:     **else**
15:         STATE $\leftarrow$ SPARSE

---

We give our algorithm in full in Algorithm 1. Because DETHH is a deterministic algorithm, it will always be correct in the case where the universe size is small. Thus, we first prove that in the case where the universe size is large, then ROBUSTCS ensures correctness within each sequence of $\ell$ updates.

**Lemma 2.5.** *Suppose the number of distinct elements at the beginning of a block is at least $50t$. Let $S$ be the output of* ROBUSTCS *at the beginning of a block. Then conditioned on the correctness of* ROBUSTCS*, $S$ solves the $L_p$-heavy hitter problem on the entire block.*

Next, we show that ROBUSTCS ensures correctness in between blocks as well. We also analyze the space complexity of our algorithm.

**Lemma 2.6.** *With high probability,* ROBUSTCS *is correct at the beginning of each block of length $\ell$.*

**Lemma 2.7.** *The total space by the algorithm is $\tilde{\mathcal{O}}\left(\frac{1}{\varepsilon^{2.5}} m^{(2p-2)/(4p-3)}\right)$ bits of space.*

Given our proof of correctness in Lemma 2.5 and Lemma 2.6, as well as the space analysis in Lemma 2.7, then we obtain Theorem 1.2.

## 3 Oblivious Residual Estimation Algorithm

In this section, we consider norm and moment estimation of a residual vector, permitting bicriteria error by allowing some slack in the size of the tail. Specifically, suppose the input vector $f$ arrives in the streaming model. Given a tail parameter $k > 0$ and an error parameter $\varepsilon \in (0,1)$, let $g$ be a tail vector of $f$ that omits the $k$ entries of $f$ largest in magnitude, breaking ties arbitrarily and let $h$ be a tail vector of $f$ that omits the $(1-\varepsilon)k$ entries of $f$ largest in magnitude. We give an algorithm that estimates $\|g\|_p^p$ up to additive $\varepsilon \cdot \|h\|_p^p$, using space poly $\left(\frac{1}{\varepsilon}, \log n\right)$, which is independent of the tail parameter $k$. It should be noted that our algorithm is imprecise on $\|g\|_p^p$ in two ways. Firstly, it incurs additive error proportional to $\varepsilon$. Secondly, the additive error has error with respect to $h$, which is missing the top $(1-\varepsilon)k$ entries of $f$ in magnitude, rather than the top $k$. Nevertheless, the space bounds that are independent of $k$ are sufficiently useful for our subsequent application of $L_p$ estimation. We first define the level sets of the $p$-th moment and the contribution of each level set.

**Definition 3.1** (Level sets and contribution). *Let $\eta > 0$ be a parameter and let $m$ be the length of the stream. Let $M$ be the power of two such that $m^p \leq M < (1+\eta)m^p$ and let $\zeta \in [1, 2]$. Then for each integer $\ell \geq 1$, we define the level set $\Gamma_\ell := \left\{ i \in [n] \mid f_i \in \left[ \frac{\zeta M}{(1+\eta)^{\ell-1}}, \frac{\zeta M}{(1+\eta)^\ell} \right) \right\}$. We also define the contribution $C_\ell$ of level set $\Gamma_\ell$ to be $C_\ell := \sum_{i \in \Gamma_\ell} (f_i)^p$.*

For a residual vector $g$ of $f$ with the top $k$ coordinates set to be zero, we similarly define the level sets $\Lambda_\ell$ and $D_\ell$ of $g$ in the natural way, i.e., $D_\ell := \sum_{i \in \Lambda_\ell} (g_i)^p$ for $\Lambda_\ell := \left\{ i \in [n] \mid g_i \in \left[ \frac{\zeta M}{(1+\eta)^{\ell-1}}, \frac{\zeta M}{(1+\eta)^\ell} \right) \right\}$.

---

**Algorithm 2** RESIDUALEST: residual $F_p$ approximation algorithm, $p \in [1, 2]$

---

**Input:** Stream $s_1, \ldots, s_m$ of items from $[n]$, accuracy parameter $\varepsilon \in (0,1)$, $p \in [1,2]$
**Output:** $(1 + \varepsilon)$-approximation to $F_p$

1: $\eta \leftarrow \frac{\varepsilon}{100}$, $L \leftarrow \tilde{\mathcal{O}}\left(\frac{\log(nm)}{\eta}\right)$, $P = \tilde{\mathcal{O}}\left(\log(nm)\right)$, $R \leftarrow \tilde{\mathcal{O}}\left(\log \frac{\log n}{\eta}\right)$, $\gamma \leftarrow 2^{20}$
2: **for** $t = 1$ to $t = m$ **do**
3:     **for** $(i, r) \in [P] \times [R]$ **do**
4:         Let $U_i^{(r)}$ be a (nested) subset of $[n]$ subsampled at rate $p_i := \min(1, 2^{1-i})$
5:         **if** $s_t \in U_i^{(r)}$ **then**
6:             Send $s_t$ to COUNTSKETCH$_i^{(r)}$ with accuracy $\eta^3$
7: Let $M = 2^i$ for some integer $i \geq 0$, such that $m^p \leq M < 2m^p$
8: $c \leftarrow k$
9: **for** $\ell = 1$ to $\ell = L$ **do**
10:     $i \leftarrow \max\left(1, \left\lfloor \log(1+\eta)^\ell - \log \frac{\gamma^2 \log(nm)}{\eta^3} \right\rfloor\right)$
11:     Let $H_i^{(r)}$ be the outputs of COUNTSKETCH at level $i$
12:     Let $S_i^{(r)}$ be the set of ordered pairs $(j, \widehat{f_j})$ of $H_i^{(r)}$ with $\left(\widehat{f_j}\right)^p \in \left[\frac{\zeta M}{(1+\eta)^{\ell-1}}, \frac{\zeta M}{(1+\eta)^\ell}\right]$
13:     $\widehat{|\Gamma_\ell|} \leftarrow \frac{1}{p_i} \text{median}_{r \in [R]} |S_i^{(r)}|$, $T_\ell \leftarrow \max(0, \widehat{\Gamma_\ell} - c)$
14:     $c \leftarrow \max(c - \widehat{\Gamma_\ell}, 0)$
15:     $\widehat{|\Lambda_\ell|} \leftarrow T_\ell \cdot (1 + \eta)^\ell$
16: Return $\widehat{F_{p, \text{Res}(k)}} = \sum_{\ell \in [L]} \widehat{|\Lambda_\ell|}(1 + \eta)^\ell$

---

Our algorithm attempts to estimate the contribution of each level set. Some of these level sets contribute a "significant" amount to the $p$-th moment of $f$, whereas other level sets do not. It can be seen that the number of items in each level set that is contributing can be estimated up to a $(1 + \mathcal{O}(\varepsilon))$-approximation. In particular, either a contributing level set has a small number of items with large mass, or a large number of items that collectively have significant mass. We use the heavy-hitter algorithm COUNTSKETCH to detect the level sets with a small number of items with large mass, and count the number of items in these level sets. For the large number of items that collectively have significant mass, it can be shown that after subsampling the universe, there will be a large number of these items remaining, and those items will be identified by COUNTSKETCH on the subsampled universe. Moreover, the total number of such items in the level set can be estimated accurately by rescaling the number of the heavy-hitters in the subsampled universe inversely by the sampling probability. We can thus carefully count the number of items in the contributing level sets and subtract off the largest $k$ such items. Because we only have $(1 + \varepsilon)$-approximations to the number of such items, it may be possible that we subtract off too many, hence the bicriteria approximation.

Finally, we note that for the insignificant level sets, we can no longer estimate the number of items in these level set up to $(1 + \varepsilon)$-factor. However, we note that the number of such items is only an $\varepsilon$ fraction of the number of items in the lower level sets that are contributing. Therefore, we can show that it suffices to set the contribution of these level sets to zero. Our algorithm appears in full in Algorithm 2.

We now show that the number of items (as well as their contribution) in each "contributing" level set with a small number of items with large mass will be estimated within a $(1 + \varepsilon)$-approximation.

**Lemma 3.2.** *Let $r \in [R]$ be fixed. Then with probability at least $\frac{9}{10}$, we have that simultaneously for all $j \in U_i^{(r)}$ for which $(f_j)^p \geq \frac{\eta^3 \cdot F_p(U_i^{(r)})}{2^7 \gamma \log^2(nm)}$, $H_\ell^{(r)}$ outputs $\widehat{f_j}$ with $\left(1 - \frac{\eta}{8 \log(nm)}\right) \cdot (f_j)^p \leq (\widetilde{f_j})^p \leq \left(1 + \frac{\eta}{8 \log(nm)}\right) \cdot (f_j)^p$.*

We now show that the number of items in each "contributing" level set is estimated within a $(1 + \varepsilon)$-approximation, including the level sets that contain a large number of small items.

**Lemma 3.3.** *Given a fixed $\varepsilon \in (0, 1)$, let $\Lambda_\ell$ be a fixed level set and let $r \in [R]$ be fixed. Let $i = \max\left(1, \left\lfloor \log(1 + \eta)^\ell - \log \frac{\gamma^2 \log(nm)}{\eta^3} \right\rfloor\right)$. Define the events $\mathcal{E}_1$ to be the event that $|U_i^{(r)}| \leq \frac{32n}{2^i}$ and $\mathcal{E}_2$ to be the event that $F_p(U_i^{(r)}) \leq \frac{32 F_p}{2^i}$. Then conditioned on $\mathcal{E}_1$ and $\mathcal{E}_2$, for each $j \in \Lambda_\ell \cap U_i^{(r)}$, there exists $(j, \widetilde{f}_j)$ in $S_i^{(r)}$ such that with probability at least $\frac{9}{10}$, $\left(1 - \frac{\eta}{8 \log(nm)}\right) \cdot (f_j)^p \leq (\widetilde{f}_j)^p \leq \left(1 + \frac{\eta}{8 \log(nm)}\right) \cdot (f_j)^p$.*

Putting things together, we have the following full guarantees for our algorithm.

**Theorem 3.4.** *There exists a one-pass streaming algorithm RESIDUALEST that takes an input parameter $k \geq 0$ (possibly upon post-processing the stream) and uses $\tilde{\mathcal{O}}\left(\frac{1}{\varepsilon^6} \cdot \log^3(nm)\right)$ bits of space to output an estimate $\widehat{F_{p,\text{Res}(k)}}$ with $\mathbf{Pr}\left[\left|\widehat{F_{p,\text{Res}(k)}} - F_{p,\text{Res}(k)}\right| \leq \varepsilon \cdot F_{p,\text{Res}((1-\varepsilon)k)}\right] \geq \frac{2}{3}$.*

## 4  Adversarially Robust $L_p$ Estimation

In this section, we give an adversarially robust algorithm for $F_p$ moment estimation on turnstile streams. Due to the relationship between the $F_p$ moment and the $L_p$ norm, our result similarly translates to a robust algorithm for $L_p$ norm estimation. We first require an algorithm to recover all the coordinates of the underlying frequency vector if it is sparse.

**Theorem 4.1.** *[GSTV07] There exists a deterministic algorithm SPARSERECOVER that recovers a $k$-sparse frequency vector defined by an insertion-deletion stream of length $n$. The algorithm uses $k \cdot \text{polylog}(n)$ bits of space.*

---

**Algorithm 3** Adversarially robust $L_p$-estimation

**Input:** Turnstile stream of length $m$ for a frequency vector of dimension $n$
**Output:** Adversarially robust heavy-hitters
1: $c \leftarrow \frac{24p^2 - 23p + 4}{(4p-3)(12p+3)}$, $\gamma \leftarrow \frac{2c}{5} - \frac{(4p-4)}{(20p-15)}$, $\eta \leftarrow \frac{\varepsilon^2}{100m^\gamma}$, $k \leftarrow \mathcal{O}\left(\frac{1}{\eta^p}\right)$, $\ell \leftarrow \mathcal{O}\left(\varepsilon \cdot m^{c/p} k^{1-1/p}\right)$, STATE $\leftarrow$ SPARSE
2:   Initialize SPARSERECOVER with sparsity $\mathcal{O}(m^c)$
3:   Initialize ROBUSTHH with threshold $\varepsilon\eta$
4:   Initialize LZEROEST with accuracy 2 robust to $b := \frac{m}{\ell}$ queries
5:   Initialize RESIDUALEST with parameter $k$ and accuracy $\mathcal{O}(\varepsilon)$ robust to $b$ queries
6:   **for** each block of $\ell$ updates **do**
7:       Update ROBUSTHH, LZEROEST, and RESIDUALEST
8:       **if** STATE = SPARSE at the beginning of the block **then**
9:           Let $g$ be the vector output by SPARSERECOVER
10:          $\widehat{G} \leftarrow \|g\|_p^p$
11:          Return $\widehat{G}$
12:      **else**
13:          Let $g$ be the vector output by ROBUSTHH at the beginning of the block
14:          Let $\widehat{H}$ be the output of RESIDUALEST
15:          $\widehat{G} \leftarrow \|g\|_p^p$
16:          Return $\widehat{G} + \widehat{H}$
17:      Let $Z$ be the output of robust LZEROEST
18:      **if** $Z > 100t$ **then**
19:          STATE $\leftarrow$ DENSE
20:      **else**
21:          STATE $\leftarrow$ SPARSE

---

We remark that SPARSERECOVER is deterministic and guarantees correctness on a turnstile stream, even if the frequency vector is not sparse at some intermediate step of the stream. On the other hand,

if the frequency vector is not sparse, then a query to SPARSERECOVER could be erroneous. Hence, our algorithm thus utilizes robust LZEROEST to detect whether the underlying frequency vector is dense or sparse. Similar to [BEO22], the intuition is that due to the sparse case always succeeding, the adversary can only induce failure if the vector is dense, which in turn decreases the flip number. However, because we also accurately track the heavy-hitters, then the adversary must spread the updates across a multiple number of coordinates, resulting in a larger number of updates necessary to double the residual vector. Since the number of updates is larger, then the flip number is smaller, and so our algorithm can use less space. Unfortunately, even though the residual vector may not double in its $p$-th moment, the $p$-th moment of entire frequency vector $f$ may change drastically. This is a nuance for the analysis because our error guarantee can no longer be relative to the $\|f\|_p^p$. Indeed, $\varepsilon \cdot \|f\|_p^p$ additive error may induce $(1+\varepsilon)$-multiplicative error at one point, but at some later point we could have $\|f'\|_p^p \ll \|f\|_p^p$, so that the same additive error could even be polynomial multiplicative error. Hence, we require the RESIDUALEST subroutine from Section 3, whose guarantees are in terms of the residual vector. We give our algorithm in full in Algorithm 3.

We upper bound the amount that the $p$-th moment of the residual vector can change, given a bounded number of updates.

**Lemma 4.2.** *Let $f$ be a frequency vector and $g$ be the residual vector omitting the $k$ coordinates of $f$ largest in magnitude. Let $v$ be any arbitrary vector such that $\|v\|_1 \le \frac{\varepsilon}{100} \cdot \|g\|_p \cdot k^{1-1/p}$ and $\|v\|_1 \le \frac{1}{2}\|g\|_1$. Let $u$ be the residual vector omitting the $k$ coordinates of $f + v$ largest in magnitude. Then we have $|\|g\|_p^p - \|u\|_p^p| \le \frac{\varepsilon}{4} \cdot \|g\|_p^p$.*

We now show correctness and space complexity of Algorithm 3, after which Theorem 1.3 follows.

**Lemma 4.3.** *For $\log n = \Theta(\log m)$, Algorithm 3 uses $\tilde{\mathcal{O}}\left(\frac{1}{\varepsilon^{7.5}} \cdot m^c\right)$ bits of space in total. Moreover, for any fixed time during a stream, let $f$ be the induced frequency vector and let $\widehat{F}$ be the output of Algorithm 3. Then we have that with high probability, $(1-\varepsilon)\|f\|_p^p \le \widehat{F} \le (1+\varepsilon)\|f\|_p^p$.*

## 5   Empirical Evaluations

In this section, we describe our empirical evaluations for comparing the flip number of the entire vector and the flip number of the residual vector on real-world datasets. Note that these quantities parameterize the space used by the algorithm of [BEO22] and by our algorithm, respectively.

**CAIDA traffic monitoring dataset.** We used the CAIDA dataset [CAI16] of anonymized passive traffic traces from the 'equinix-nyc' data center's high-speed monitor. The dataset is commonly used for empirical evaluations on frequency moments and heavy-hitters. We extracted the sender IP addresses from 12 minutes of the internet flow data, which contained 2,9922,873 total events.

**Experimental setup.** Our empirical evaluations were performed Python 3.10 on a 64-bit operating system on an AMD Ryzen 7 5700U CPU, with 8GB RAM and 8 cores with base clock 1.80 GHz. We compare the flip number of the entire data stream versus the flip number of the residual vector across various values of the algorithm error $\varepsilon \in \{10^{-1}, 10^{-2}, \ldots, 10^{-5}\}$, values of the heavy-hitter threshold $\alpha \in \{4^{-1}, 4^{-2}, \ldots, 4^{-10}\}$, and the frequency moment parameter $p \in \{1.1, 1.2, \ldots, 1.9\}$. We describe the results in Figure 1.

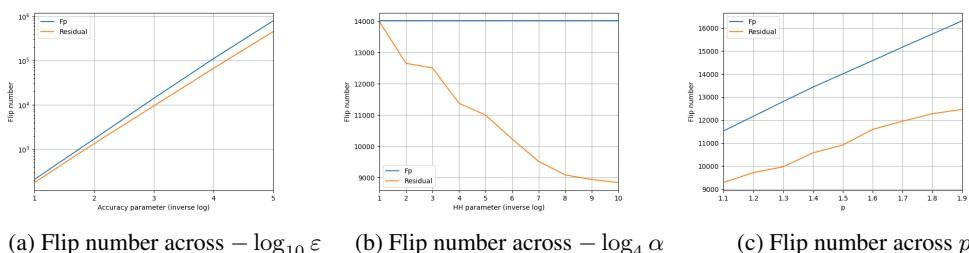

(a) Flip number across $-\log_{10} \varepsilon$     (b) Flip number across $-\log_4 \alpha$     (c) Flip number across $p$

Fig. 1: Empirical evaluations on the CAIDA dataset, comparing flip number of the $p$-th frequency moment and the residual, for $\varepsilon = \alpha = 0.001$ and $p = 1.5$ when not variable. Smaller flip numbers indicate less space needed by the algorithm.

**Results and discussion.** Our empirical evaluations serve as a simple proof-of-concept demonstrating that adversarially robust algorithm can use significantly less space than existing algorithms. In particular, existing algorithms use space that is an increasing function of the flip number of the $p$-th frequency moment, while our algorithms use space that is an increasing function of the flip number of the residual, which is significantly less across all settings in Figure 1. While the ratio does increase as the exponent $p$ increases in Figure 1c, there is not a substantial increase, i.e., 1.24 to 1.31 from $p = 1.1$ to $p = 1.9$. On the other hand, as $\alpha$ decreases in Figure 1b, the ratio increases from 1.002 for $\alpha = 4^{-1}$ to 1.6 for $\alpha = 4^{-10}$. Similarly, in Figure 1a, the ratio of these quantities begins at 1.17 for $\varepsilon = 10^{-1}$ and increases to as large as 1.75 for $\varepsilon = 10^{-5}$. Therefore, even in the case where the input is not adaptive, our empirical evaluations demonstrate that these flip number quantities can be quite different, and consequently, our algorithm can use significantly less space than previous existing algorithms.

## Acknowledgements

David P. Woodruff was supported in part by a Simons Investigator Award and NSF CCF-2335412. Samson Zhou is supported in part by NSF CCF-2335411. The work was conducted in part while David P. Woodruff and Samson Zhou were visiting the Simons Institute for the Theory of Computing as part of the Sublinear Algorithms program.

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

## A   Preliminaries

For a positive integer $n > 0$, we use $[n]$ to denote the set of integers $\{1, \ldots, n\}$. We use $\mathrm{poly}(n)$ to denote a fixed polynomial in $n$ whose degree can be set by adjust constants in the algorithm based on various desiderata, e.g., in the failure probability. We use $\mathrm{polylog}(n)$ to denote $\mathrm{poly}(\log n)$. When there exist constants to facilitate an event to occur with probability $1 - \frac{1}{\mathrm{poly}(n)}$, we say that the event occurs with high probability. For a random variable $X$, we use $\mathbb{E}[X]$ to denote its expectation and $\mathrm{Var}(X)$ to denote its variance.

Recall that for $p > 0$, the $L_p$ norm of a vector $v \in \mathbb{R}^n$ is $\|v\|_p = (v_1^p + \ldots + v_n^p)^{1/p}$. The $p$-th moment of $v$ is defined as $F_p(v) = \|v\|_p^p$. Note that for a constant $p \geq 1$, a $(1 + \varepsilon)$-approximation to the $F_p(v)$ implies a $(1 + \varepsilon)$-approximation to $\|v\|_p$. Similarly, for a sufficiently small constant $\varepsilon \in (0, 1)$, a $(1 + \mathcal{O}(\varepsilon))$-approximation to $\|v\|_p$ implies a $(1 + \mathcal{O}(\varepsilon))^p = (1 + \varepsilon)$-approximation to $F_p(v)$. We thus use the problems of $L_p$ norm estimation and $F_p$ moment estimation interchangeably in discussion.

We use $F_{p,\mathrm{Res}(k)}(f)$ to denote the $p$-th moment of a vector $g$ obtained by setting to zero the $k$ coordinates of $f$ largest in magnitude, breaking ties arbitrarily. We also define $\|v\|_0$ to be the number of nonzero coordinates of $v$, so that $\|v\|_0 = |\{i \in [n] \mid v_i \neq 0\}|$.

We recall the following notions regarding differential privacy.

**Definition A.1** (Differential privacy). *[DMNS06] Given $\varepsilon > 0$ and $\delta \in (0,1)$, a randomized algorithm $\mathcal{A} : D \to R$ with domain $D$ and range $R$ is $(\varepsilon, \delta)$-differentially private if, for every neighboring datasets $S$ and $S'$ and for all $\mathcal{E} \subseteq R$,*

$$\mathbf{Pr}\left[\mathcal{A}(S) \in \mathcal{E}\right] \leq e^\varepsilon \cdot \mathbf{Pr}\left[\mathcal{A}(S') \in \mathcal{E}\right] + \delta.$$

**Theorem A.2** (Private median, e.g., [HKM$^+$20]). *Given a database $\mathcal{D} \in X^*$, there exists an $(\varepsilon, 0)$-differentially private algorithm PRIVMED that outputs an element $x \in X$ such that with probability at least $1 - \delta$, there are at least $\frac{|S|}{2} - k$ elements in $S$ that are at least $x$, and at least $\frac{|S|}{2} - k$ elements in $S$ in $S$ that are at most $x$, for $k = \mathcal{O}\left(\frac{1}{\varepsilon}\log\frac{|X|}{\delta}\right)$.*

**Theorem A.3** (Advanced composition, e.g., [DRV10]). *Let $\varepsilon, \delta' \in (0,1]$ and let $\delta \in [0,1]$. Any mechanism that permits $k$ adaptive interactions with mechanisms that preserve $(\varepsilon, \delta)$-differential privacy guarantees $(\varepsilon', k\delta + \delta')$-differential privacy, where $\varepsilon' = \sqrt{2k\ln\frac{1}{\delta'}} \cdot \varepsilon + 2k\varepsilon^2$.*

**Theorem A.4** (Generalization of DP, e.g., [DFH$^+$15, BNS$^+$21]). *Let $\varepsilon \in (0, 1/3)$, $\delta \in (0, \varepsilon/4)$, and $n \geq \frac{1}{\varepsilon^2}\log\frac{2\varepsilon}{\delta}$. Suppose $\mathcal{A} : X^n \to 2^X$ is an $(\varepsilon, \delta)$-differentially private algorithm that curates a database of size $n$ and produces a function $h : X \to \{0, 1\}$. Suppose $\mathcal{D}$ is a distribution over $X$ and $S$ is a set of $n$ elements drawn independently and identically distributed from $\mathcal{D}$. Then*

$$\Pr_{S \sim \mathcal{D}, h \leftarrow \mathcal{A}(S)}\left[\left|\frac{1}{|S|}\sum_{x \in S} h(x) - \mathbb{E}_{x \sim \mathcal{D}}[h(x)]\right| \geq 10\varepsilon\right] < \frac{\delta}{\varepsilon}.$$

---

**Algorithm 4** Adversarially Robust Framework

---

**Input:** Oblivious algorithms $\mathcal{A}$ with failure probability $\delta_0$, number of queries $Q$, failure probability $\delta$

**Output:** Algorithm robust to $Q$ queries, with failure probability at most $\delta$

1: $r \leftarrow \mathcal{O}\left(\sqrt{Q}\log^2\frac{Q}{\delta\delta_0}\right)$
2: Implement $k = \mathcal{O}(r)$ independent instances $\mathcal{A}_1, \ldots, \mathcal{A}_k$ of $\mathcal{A}$ on the input
3: **for** each query $q_i$, $i \in [Q]$ **do**
4:      Let $Z_{i,j}$ be the output of $\mathcal{A}_j$ on $q_i$
5:      Let PRIVMED be $\left(\frac{1}{r}, 0\right)$-DP
6:      Return PRIVMED$\left(\{Z_{i,j}\}_{j \in [k]}\right)$

---

We remark that Algorithm 4 is the algorithm corresponding to the statement of Theorem 2.3

## B    Missing Proofs from Section 2

One reason that DETHH is not commonly utilized is that with the additional power of randomness, significantly better space bounds can be achieved, such as by the following guarantees:

**Theorem B.1.** *[CCF04] For $p \in [1, 2)$, there exists a randomized algorithm COUNTSKETCH that solves the $\varepsilon$-$L_p$ heavy-hitters on a universe of size $n$ and a stream of length $m$ and uses $\mathcal{O}\left(\frac{1}{\varepsilon^2}\log n \log\frac{nm}{\delta}\right)$ bits of space.*

To achieve the guarantees of Theorem 2.2, a natural approach would be to apply Theorem 2.3 to the guarantees of COUNTSKETCH in Theorem B.1. However, this does not achieve the optimal bounds because each round of adaptive queries can require multiple answers, i.e., estimated frequencies for each of the heavy-hitters at that time. Thus, [CLN$^+$22] proposed a slight variation of the algorithm along with intricate analysis to achieve the guarantees of Theorem 2.2.

**Lemma 2.5.** *Suppose the number of distinct elements at the beginning of a block is at least $50t$. Let $S$ be the output of ROBUSTCS at the beginning of a block. Then conditioned on the correctness of ROBUSTCS, $S$ solves the $L_p$-heavy hitter problem on the entire block.*

*Proof.* Suppose the number of distinct elements at the beginning of a block is at least $50t$. Let $f$ be the frequency vector at the beginning of the block and let $g$ be the frequency vector at any

intermediate step in the block. Conditioned on the correctness of ROBUSTCS, we have that the estimated frequency $\widehat{f}_i$ of each item $i$ satisfies

$$|\widehat{f}_i - f_i| \le \frac{t^{1/p}}{100}.$$

Thus if $f_i$ is an $\frac{\varepsilon}{2}$-$L_p$ heavy hitter, then $i \in S$ and conversely if $i \in S$, then $f_i \ge \frac{\varepsilon}{4}\|f\|_p$.

Since each block has length $\ell = \frac{\varepsilon}{100}t^{1/p}$, then $|g_i - f_i| \le \frac{\varepsilon}{100}t^{1/p}$. Moreover, because the number of distinct elements is at least $50t$, then we have $\|f\|_p \ge 50t^{1/p}$. Therefore if $g_i$ is an $\varepsilon$-$L_p$ heavy hitter, then $f_i$ is an $\frac{\varepsilon}{2}$-$L_p$ heavy hitter, so that $i \in S$. Similarly if $g_i < \frac{\varepsilon}{2}\|g\|_p$, then $f_i < \frac{\varepsilon}{4}\|f\|_p$, so that $i \notin S$. $\qquad\square$

**Lemma 2.6.** *With high probability, ROBUSTCS is correct at the beginning of each block of length $\ell$.*

*Proof.* We have $\ell = \frac{\varepsilon}{100} \cdot t^{1/p}$. Note that there are $b = \frac{m}{\ell} = \frac{100m}{\varepsilon t^{1/p}}$ blocks of length $\ell$. Thus it suffices to require ROBUSTCS to be robust to $b$ queries in the subroutine ADAPTIVEHH to achieve correctness at the beginning of each block, with high probability. $\qquad\square$

**Lemma 2.7.** *The total space by the algorithm is $\tilde{\mathcal{O}}\left(\frac{1}{\varepsilon^{2.5}}m^{(2p-2)/(4p-3)}\right)$ bits of space.*

*Proof.* Since DETHH is called with threshold $\frac{\varepsilon}{16}$ for $t = \mathcal{O}\left(m^{p/(4p-3)}\right)$, then the total space by DETHH is $\tilde{\mathcal{O}}\left(\frac{1}{\varepsilon^2}t^{2-2/p}\right) = \tilde{\mathcal{O}}\left(\frac{1}{\varepsilon^2}m^{(2p-2)/(4p-3)}\right)$ bits.

We require ROBUSTCS to be robust to $b$ queries in the subroutine ADAPTIVEHH, thus using space $\tilde{\mathcal{O}}\left(\frac{\sqrt{b}}{\varepsilon^2}\log n \log \frac{nm\lambda}{\delta}\right)$ for $\delta = \frac{1}{\text{poly}(n,m)}$ and

$$\tilde{\mathcal{O}}\left(\sqrt{b}\right) = \tilde{\mathcal{O}}\left(\sqrt{\frac{m}{\varepsilon t^{1/p}}}\right) = \tilde{\mathcal{O}}\left(\varepsilon^{-1/2}m^{(2p-2)/(4p-3)}\right).$$

Similarly, we use $\tilde{\mathcal{O}}\left(\sqrt{b}\right) = \tilde{\mathcal{O}}\left(\varepsilon^{-1/2}m^{(2p-2)/(4p-3)}\right)$ instances of LZEROEST to guarantee robustness against $b$ queries. Each instance of LZEROEST uses $\mathcal{O}\left(\frac{1}{\varepsilon^2}\log^2(nm)\right)$ bits of space. Hence, the overall space is $\tilde{\mathcal{O}}\left(\frac{1}{\varepsilon^{2.5}}m^{(2p-2)/(4p-3)}\right)$ bits. $\qquad\square$

## C  Missing Proofs from Section 3

We revisit the guarantee of COUNTSKETCH with a different parameterization in this section.

**Theorem C.1.** *[CCF04] Given $p \in [1,2]$, there exists a one-pass streaming algorithm COUNTSKETCH that with high probability, reports all $j \in [n]$ for which $(f_j)^p \ge \varepsilon^p F_p$, along with estimations $\widehat{f}_j$, such that $(1-\varepsilon)f_j^p \le (\widehat{f}_j)^p \le (1+\varepsilon)f_j^p$.*

Observe that to provide the guarantees of Theorem C.1, COUNTSKETCH would require space $\text{poly}\left(\frac{1}{\varepsilon}, \log n\right)$, rather than quadratic dependency $\frac{1}{\varepsilon}$.

**Lemma 3.2.** *Let $r \in [R]$ be fixed. Then with probability at least $\frac{9}{10}$, we have that simultaneously for all $j \in U_i^{(r)}$ for which $(f_j)^p \ge \frac{\eta^3 \cdot F_p(U_i^{(r)})}{2^7 \gamma \log^2(nm)}$, $H_\ell^{(r)}$ outputs $\widehat{f}_j$ with $\left(1 - \frac{\eta}{8\log(nm)}\right) \cdot (f_j)^p \le (\widetilde{f}_j)^p \le \left(1 + \frac{\eta}{8\log(nm)}\right) \cdot (f_j)^p$.*

*Proof.* The proof follows from Theorem C.1 and the fact that COUNTSKETCH is only run on the substream induced by $I_\ell^{(r)}$. $\qquad\square$

*Proof.* Consider casework on $\left\lfloor \log(1+\eta)^\ell - \log \frac{\gamma^2 \log(nm)}{\eta^3} \right\rfloor \le 1$ or $\left\lfloor \log(1+\eta)^\ell - \log \frac{\gamma^2 \log(nm)}{\eta^3} \right\rfloor > 1$. Informally, the casework corresponds to whether the frequencies $\left(\widehat{f}_j\right)^p$ in a significant level set are large or not large, i.e., whether they are above the

heavy-hitter threshold before subsampling the universe. Thus if the frequencies are large, then the heavy-hitter algorithm will estimate their frequencies, but if the frequencies are not large, then we must perform subsampling before the items surpass the heavy-hitter threshold.

Suppose $\left\lfloor \log(1 + \eta)^\ell - \log \frac{\gamma^2 \log(nm)}{\eta^3} \right\rfloor \leq 1$, so that $\frac{1}{(1+\eta)^{\ell-1}} \geq \frac{\eta^3}{\gamma \log^2(nm)}$. Note that $j \in \Lambda_\ell$ implies $(f_j)^p \in \left[ \frac{\zeta M}{(1+\eta)^{\ell-1}}, \frac{\zeta M}{(1+\eta)^\ell} \right)$ and thus $(f_j)^p \geq \frac{\eta^3 \zeta M}{\gamma \log^2(nm)}$. Note that $M \geq F_p$ and thus by Lemma 3.2, we have that with probability at least $\frac{9}{10}$, $H_i^{(r)}$ outputs $\widehat{f}_j$ such that

$$\left( 1 - \frac{\eta}{8 \log(nm)} \right) \cdot (f_j)^p \leq (\widetilde{f}_j)^p \leq \left( 1 + \frac{\eta}{8 \log(nm)} \right) \cdot (f_j)^p,$$

as desired.

For the other case, suppose $\left\lfloor \log(1+\eta)^\ell - \log \frac{\gamma^2 \log(nm)}{\eta^3} \right\rfloor > 1$, so that $i = \left\lfloor \log(1+\eta)^\ell - \log \frac{\gamma^2 \log(nm)}{\eta^3} \right\rfloor$. Since $p_i = 2^{1-i}$, then we have that

$$p_i = \frac{2\gamma \log^2(nm)}{(1+\eta)^\ell \eta^3}.$$

Since $j \in \Lambda_i$, we have again $(f_j)^p \in \left[ \frac{\zeta M}{(1+\eta)^{\ell-1}}, \frac{\zeta M}{(1+\eta)^\ell} \right)$ and therefore,

$$(f_j)^p \geq \frac{F_p}{4 \cdot (1+\eta)^\ell} \geq \frac{\eta^3}{4\gamma \log^2(nm)} \frac{F_p}{2^{i-1}}.$$

Conditioning on the event $\mathcal{E}_2$, we have $F_p(U_i^{(r)}) \leq \frac{32 F_p}{2^i}$ and thus

$$(f_j)^p \geq \frac{\eta^3}{4\gamma \log^2(nm)} \frac{2 F_p}{2^i} \geq \frac{\eta^3}{128 \gamma \log^2(nm)} \cdot F_p(U_i^{(r)}).$$

Hence by Lemma 3.2, we have that with probability at least $\frac{9}{10}$, $H_i^{(r)}$ outputs $\widehat{f}_j$ such that

$$\left( 1 - \frac{\eta}{8 \log(nm)} \right) \cdot (f_j)^p \leq (\widetilde{f}_j)^p \leq \left( 1 + \frac{\eta}{8 \log(nm)} \right) \cdot (f_j)^p,$$

as desired. □

We now give the correctness guarantees of Algorithm 2.

**Lemma C.2.** $\mathbf{Pr}\left[ \left| \widehat{F_{p,\mathrm{Res}(k)}} - F_{p,\mathrm{Res}(k)} \right| \leq \varepsilon \cdot F_{p,\mathrm{Res}((1-\varepsilon)k)} \right] \geq \frac{2}{3}.$

*Proof.* We would like to show that for each level set $\ell$, we accurately estimate its residual contribution $D_\ell$. More specifically, we would like to show $|\widehat{D_\ell} - D_\ell| \leq \frac{\eta}{8 \log(nm)} \cdot F_p$ for all $\ell \in [L]$. Let $g$ be the residual vector of $f$ with the largest $k$ coordinates in magnitude set to zero. For a level set $\ell$, we define the fractional contribution $\phi_\ell := \frac{C_\ell}{\sum_{i \in [n]} (f_i)^p}$. Given an accuracy parameter $\varepsilon$ and a stream of length $m$, we define a level set $\Lambda_\ell$ to be *significant* if $\phi_\ell \geq \frac{\varepsilon^2 \eta}{100 p \log(nm)}$. Furthermore, we define a level set $\Lambda_\ell$ to be *contributing* if $\phi_\ell \geq \frac{\varepsilon \eta}{100 p \log(nm)}$. Otherwise, the level set is defined to be $\phi_\ell < \frac{\varepsilon^2 \eta}{100 p \log(nm)}$.

For a fixed $\ell$, we have that $D_\ell = \sum_{j \in \Lambda_\ell} (g_j)^p$, where $j \in \Lambda_\ell$ if $(g_j)^p \in \left[ \frac{\zeta M}{(1+\eta)^{\ell-1}}, \frac{\zeta M}{(1+\eta)^\ell} \right)$. On the other hand, for each fixed $r$, we have that $S_\ell^{(r)}$ is determined using items $j$ whose estimated frequency are in the range $(\widehat{g}_j)^p \in \left[ \frac{\zeta M}{(1+\eta)^{\ell-1}}, \frac{\zeta M}{(1+\eta)^\ell} \right)$, so it is possible that $j$ could be classified into contributing to $\Lambda_\ell$ even if $j \notin \Lambda_\ell$. Hence, we first analyze an "idealized" setting, where each index $j$ is correctly classified across all level sets $\ell \in [L]$. We that we achieve a $(1 + \widetilde{\mathcal{O}}(\varepsilon))$-approximation to $F_p$ in the idealized setting and then argue that because we choose $\zeta$ uniformly at random, then only approximation guarantee will worsen only slightly but still remain a $(1 + \varepsilon)$-approximation to $F_p$, since only a small number of coordinates will be misclassified and so our approximation guarantee will only slightly degrade.

**Idealized setting.** For a fixed $r \in [R]$, let $\mathcal{E}_1$ be the event that $|U_i^{(r)}| \leq \frac{32n}{2^\ell}$ and let $\mathcal{E}_2$ be the event that $F_p(U_i^{(r)}) \leq \frac{32F_p}{2^\ell}$. Note that $M \geq F_p$ and thus conditioned on $\mathcal{E}_1, \mathcal{E}_2$, then by Lemma 3.2, we have that with probability at least $\frac{9}{10}$, $H_i^{(r)}$ outputs $\widehat{f}_j$ such that

$$\left(1 - \frac{\eta}{8\log(nm)}\right) \cdot (f_j)^p \leq (\widetilde{f}_j)^p \leq \left(1 + \frac{\eta}{8\log(nm)}\right) \cdot (f_j)^p,$$

as desired.

We first show that when $(\widehat{f}_j)^p$ is correctly classified for all $j$ into level sets $\ell \in [L]$, then with probability $1 - \frac{1}{\text{poly}(nm)}$, we have that simultaneously for each fixed level set $\ell$, $|\widehat{D}_\ell - D_\ell| \leq \frac{\eta}{8\log(nm)} \cdot F_p$.

We define $\widehat{D}_\ell = T_\ell \cdot (1 + \eta)^\ell$, where $T_\ell$ is the estimated size of $\Lambda_\ell$, formed by attempting to truncate the top $k$ coordinates across the level sets $\Gamma_\ell$. In particular, we define $\widehat{|\Gamma_\ell|} = \frac{1}{p_i} \text{median}_{r \in [R]} |S_i^{(r)}|$ for $i = \max\left(1, \left\lfloor \log(1+\eta)^\ell - \log \frac{\gamma^2 \log(nm)}{\eta^3} \right\rfloor\right)$.

We analyze the expectation and variance of $\widehat{|\Gamma_\ell|}$. Firstly, let $r \in [R]$ be fixed and for each $j \in \Gamma_\ell$, let $Y_j$ be the indicator variable for whether $Y_j \in S_i^{(r)}$. We have

$$\mathbb{E}\left[\frac{1}{p_i} \cdot |\Gamma_\ell \cap S_i^{(r)}|\right] = \frac{1}{p_i} \cdot \sum_{j \in \Gamma_\ell} \mathbb{E}[Y_j] = \frac{1}{p_i} \cdot (p_i \cdot |\Gamma_\ell|) = |\Gamma_\ell|.$$

Similarly, we have

$$\text{Var}\left(\frac{1}{p_i} \cdot |\Gamma_\ell \cap S_i^{(r)}|\right) \leq \frac{1}{p_i^2} \cdot \sum_{j \in \Gamma_\ell} \mathbb{E}[Y_j]$$

$$= \frac{1}{p_i^2} \cdot (p_i \cdot |\Gamma_\ell|) = \frac{|\Gamma_\ell|}{p_i}.$$

Because $p_i \geq \min\left(1, \frac{\gamma^2 \log^2(nm)}{(1+\eta)^\ell \eta^3}\right)$, then we have that by Chebyshev's inequality,

$$\mathbf{Pr}\left[\left|\frac{1}{p_i} \cdot |\Gamma_\ell \cap S_i^{(r)}| - |\Gamma_\ell|\right| \geq\right] |\Gamma_\ell| \cdot \sqrt{(1+\eta)^\ell \eta^3} \leq \frac{1}{10},$$

conditioned on the events $\mathcal{E}_1, \mathcal{E}_2$, and $\mathcal{E}_3$.

To analyze the probability of the events $\mathcal{E}_1$ and $\mathcal{E}_2$, recall that in $U_i^{(r)}$, each item is sampled with probability $2^{-i+1}$. Hence,

$$\mathbb{E}\left[|U_i^{(r)}|\right] \leq \frac{n}{2^{i-1}}, \qquad \mathbb{E}\left[F_p(U_i^{(r)})\right] \leq \frac{F_p}{2^{i-1}}.$$

We define $\mathcal{E}_1$ to be the event that $|U_i^{(r)}| \leq \frac{32n}{2^i}$. By Markov's inequality, we have $\mathbf{Pr}[E_1] \geq \frac{15}{16}$. Similarly, we define $\mathcal{E}_2$ to be the event that $F_p(U_i^{(r)}) \leq \frac{32F_p}{2^i}$. By Markov's inequality, we also have $\mathbf{Pr}[E_2] \geq \frac{15}{16}$. We have that $\Pr \mathcal{E}_3 \mid \mathcal{E}_1 \wedge \mathcal{E}_2 \geq \frac{9}{10}$. Thus by a union bound,

$$\mathbf{Pr}\left[\left|\frac{1}{p_i} \cdot |\Gamma_\ell \cap S_i^{(r)}| - |\Gamma_\ell|\right| \geq\right] |\Gamma_\ell| \cdot \sqrt{(1+\eta)^\ell \eta^3} \leq \frac{1}{3}.$$

By Chernoff bounds, we thus have

$$\mathbf{Pr}\left[\left|\widehat{|\Gamma_\ell|} - |\Gamma_\ell|\right| \geq\right] |\Gamma_\ell| \cdot \sqrt{(1+\eta)^\ell \eta^3} \leq \text{poly}\left(\frac{\varepsilon}{\log(nm)}\right).$$

Moreover, if level $\ell$ is significant, then either $p_i = 0$ or $|\Gamma_\ell| \geq \frac{(1+\eta)^\ell}{2\eta^3}$. If $p_i = 0$, then $\widehat{|\Gamma_\ell|} = |\Gamma_\ell|$. Otherwise if $|\Gamma_\ell| \geq \frac{(1+\eta)^\ell}{2\eta^3}$, then with probability at least $1 - \text{poly}\left(\frac{\varepsilon}{\log(nm)}\right)$, we have that simultaneously for all significant levels $\ell \in [L]$, $(1-\eta)|\Gamma_\ell| \leq \widehat{|\Gamma_\ell|} \leq (1+\eta)|\Gamma_\ell|$ or in other words,

$$\left|\widehat{|\Gamma_\ell|} - |\Gamma_\ell|\right| \leq \eta|\Gamma_\ell|.$$

Since we subtract off the top $k$ coordinates in $\Gamma_\ell$ to form $\widehat{|\Lambda_\ell|}$ then we also have $\widehat{|\Lambda_\ell|} - \Lambda_\ell \leq \eta|\Gamma_\ell|$. It follows that since $j \in \Lambda_\ell$ for $(g_j)^p \in \left[\frac{\zeta M}{(1+\eta)^{\ell-1}}, \frac{\zeta M}{(1+\eta)^\ell}\right)$, then for $\widehat{D_\ell} = |\Lambda_\ell| \cdot (1+\eta)^\ell$, we have that $|\widehat{D_\ell} - D_\ell| \leq \eta(1+\eta)C_\ell$. Taking the sum over all the significant levels, we see that the error is at most $\sum_{\ell \in [L]} 2\eta C_\ell \leq \frac{\varepsilon^2}{2} \cdot F_{p,\mathrm{Res}((1-\varepsilon)k)}$.

Note that the same guarantee holds if level $\ell$ is insignificant, provided that $|\Gamma_\ell| < \frac{(1+\eta)^\ell}{1000\eta^3}$. On the other hand, if level $\ell$ is insignificant and $|\Gamma_\ell| < \frac{(1+\eta)^\ell}{1000\eta^3}$. Thus with probability at least $1 - \mathrm{poly}\left(\frac{\varepsilon}{\log(nm)}\right)$, we have that simultaneously for all insignificant levels $\ell \in [L]$, [

$$\widehat{|\Gamma_\ell|} \leq \frac{1}{200\eta^3}.$$

Then we set $\widehat{D_\ell} = 0$, so that $|\widehat{D_\ell} - D_\ell| = D_\ell$. In fact, we observe that the number of items in insignificant level sets can only be at most an $\eta$ fraction of the items in the contributing level sets beneath them. Since the sum of the contributions of contributing level sets is at most $F_{p,\mathrm{Res}((1-\varepsilon)k)}$, then taking the sum over all the significant levels, we see that the error is at most the contribution of the tail of the insignificant levels, which by definition is at most $\frac{\varepsilon}{2} \cdot F_{p,\mathrm{Res}((1-\varepsilon)k)}$.

**Randomized boundaries.** By Lemma 3.2, we have that conditioned on $\mathcal{E}_3$,

$$\left(1 - \frac{\eta}{8\log(nm)}\right) \cdot (f_j)^p \leq (\widetilde{f}_j)^p \leq \left(1 + \frac{\eta}{8\log(nm)}\right) \cdot (f_j)^p,$$

independently of the choice of $\zeta$. Because we drawn $\zeta \in [1,2]$ uniformly at random, then the probability that $j \in [n]$ is misclassified is at most $\frac{\eta}{2\log(nm)}$.

If $j \in [n]$ is indeed misclassified, then it can only be classified into either level set $\Gamma_{\ell+1}$ or level set $\Gamma_{\ell-1}$, since $\left(\widehat{f_j}^p\right)$ is a $\left(1 + \frac{\eta}{8\log(nm)}\right)$-approximation to $(f_j)^p$. As a result, a misclassified index induces at most $\eta(f_j)^p$ additive error to the contribution of level set $\Gamma_\ell$ and hence at most $\eta(f_j)^p$ additive error to the contribution of level set $\Lambda_\ell$ in the residual vector. Therefore, the total additive error across all $j \in [n]$ due to misclassification is at most $\eta \cdot F_p$ in expectation. By Markov's inequality, the total additive error due to misclassification is at most $\frac{\varepsilon}{2} \cdot F_{p,\mathrm{Res}((1-\varepsilon)k)}$ with probability at least 0.95. $\qquad\square$

**Theorem 3.4.** *There exists a one-pass streaming algorithm* RESIDUALEST *that takes an input parameter $k \geq 0$ (possibly upon post-processing the stream) and uses $\tilde{\mathcal{O}}\left(\frac{1}{\varepsilon^6} \cdot \log^3(nm)\right)$ bits of space to output an estimate $\widehat{F_{p,\mathrm{Res}(k)}}$ with $\mathbf{Pr}\left[\left|\widehat{F_{p,\mathrm{Res}(k)}} - F_{p,\mathrm{Res}(k)}\right| \leq \varepsilon \cdot F_{p,\mathrm{Res}((1-\varepsilon)k)}\right] \geq \frac{2}{3}$.*

*Proof.* Consider Algorithm 1. By Lemma C.2, we have that

$$\mathbf{Pr}\left[\left|\widehat{F_{p,\mathrm{Res}(k)}} - F_{p,\mathrm{Res}(k)}\right| \leq \varepsilon \cdot F_{p,\mathrm{Res}((1-\varepsilon)k)}\right] \geq \frac{2}{3}.$$

It thus remains to analyze the space complexity.

Note that Algorithm 1 implements $P \cdot R$ instances of COUNTSKETCH with accuracy $\eta^3$, for $P = \tilde{\mathcal{O}}(\log(nm))$, $R = \tilde{\mathcal{O}}\left(\log \frac{\log n}{\eta}\right)$, and $\eta = \frac{\varepsilon}{100}$. By Theorem C.1, each instance of COUNTSKETCH with threshold $\eta^3$ uses $\mathcal{O}\left(\frac{1}{\eta^6} \cdot \log^2(nm)\right)$ bits of space. Therefore, the total space usage of Algorithm 1 is $\tilde{\mathcal{O}}\left(\frac{1}{\varepsilon^6} \cdot \log^3(nm)\right)$ bits. $\qquad\square$

# D   Missing Proofs from Section 4

We first show that the $p$-th moment of $f$ can be essentially split by looking at the $p$-th moment of the vector consisting of the largest $k$ coordinates and the remaining tail vector.

**Lemma D.1.** *Let $\varepsilon \in (0,1)$ be a fixed accuracy parameter and let $p > 0$ be fixed. Let $f \in \mathbb{R}^n$ be any fixed vector and let $k \geq 0$ be any fixed parameter. Let $g$ be the vector consisting of the $k$ coordinates of $f$ largest in magnitude and let $h$ be the residual vector, so that $f = g + h$. Suppose $\widehat{G}$ and $\widehat{H}$ satisfy*

$$\|g\|_p^p - \frac{\varepsilon}{4}\|f\|_p^p \leq \widehat{G} \leq \|g\|_p^p + \frac{\varepsilon}{4}\|f\|_p^p$$

$$\|h\|_p^p - \frac{\varepsilon}{4}\|f\|_p^p \leq \widehat{H} \leq \|h\|_p^p + \frac{\varepsilon}{4}\|f\|_p^p.$$

*Then*

$$\left(1 - \frac{\varepsilon}{2}\right)\|f\|_p^p \leq \widehat{G} + \widehat{H} \leq \left(1 + \frac{\varepsilon}{2}\right)\|f\|_p^p.$$

*Proof.* The claim follows immediately from the fact that $\|f\|_p^p = \|g\|_p^p + \|h\|_p^p$ since $f = g + h$ but $g$ and $h$ have disjoint support. $\qquad\square$

In fact, we show the estimation is relatively accurate even if the tail vector does not quite truncate the $k$ entries largest in magnitude.

**Lemma D.2.** *Let $f$ be a frequency vector, $g$ be the residual vector omitting the $k$ coordinates of $f$ largest in magnitude, and $h$ be the residual vector omitting the $\left(\left(1 - \frac{\varepsilon}{4}\right)k\right)$ coordinates of $f$ largest in magnitude. Then we have $|\|g\|_p^p - \|h\|_p^p| \leq \frac{\varepsilon}{4} \cdot \|f\|_p^p$.*

*Proof.* Note that the smallest $\frac{\varepsilon}{4}$ coordinates of the top $k$ coordinates is only nonzero when $k \geq \frac{4}{\varepsilon}$. Thus they can only contribute $\frac{\varepsilon}{4}$ fraction to the entire moment. It follows that $|\|g\|_p^p - \|h\|_p^p| \leq \frac{\varepsilon}{4}\cdot\|f\|_p^p$, as desired. $\qquad\square$

**Lemma 4.2.** *Let $f$ be a frequency vector and $g$ be the residual vector omitting the $k$ coordinates of $f$ largest in magnitude. Let $v$ be any arbitrary vector such that $\|v\|_1 \leq \frac{\varepsilon}{100} \cdot \|g\|_p \cdot k^{1-1/p}$ and $\|v\|_1 \leq \frac{1}{2}\|g\|_1$. Let $u$ be the residual vector omitting the $k$ coordinates of $f + v$ largest in magnitude. Then we have $|\|g\|_p^p - \|u\|_p^p| \leq \frac{\varepsilon}{4} \cdot \|g\|_p^p$.*

*Proof.* Let $\|g\|_p^p = M$. Since $\|v\|_1 \leq \frac{1}{2}\|g\|_1$, then by an averaging argument $|u_i|$ can be at most $\left(\frac{8M}{k}\right)^{1/p}$ before $i$ is in the top $k$ coordinates of $f + v$. Similarly, if $i \in [n]$ is in the top $k$ coordinates of $f$ for $|v_i|$ less than $\left(\frac{8M}{k}\right)^{1/p}$, and $i$ is no longer in the top $k$ coordinates of $f + v$, then we must have $|u_i| \leq \left(\frac{16M}{k}\right)^{1/p}$. Otherwise by an averaging argument, $|u_i|$ would be too large and $i$ would be in the top $k$ coordinates of $f + v$.

Thus the contribution to $|\|g\|_p^p - \|u\|_p^p|$ is at most the contribution in the case where the number of coordinates $i$ with $|v_i| = \left(\frac{8M}{k}\right)^{1/p}$ is maximized. Because $\|v\|_1 \leq \frac{\varepsilon}{100} \cdot M \cdot k^{1-1/p}$, then there can be at most $\frac{\varepsilon}{100} \cdot k$ coordinates $i \in [n]$ such that $|v_i| \geq \left(\frac{8M}{k}\right)^{1/p}$. By the above argument, for each $i$, we have $||g_i|^p - |u_i|^p| \leq \frac{16M}{k}$. Since there can be at most $\frac{\varepsilon}{100} \cdot k$ such coordinates, then the total change in the $p$-th moment of residual is at most $16M \cdot \frac{\varepsilon}{100} \cdot k \leq \frac{\varepsilon}{4} \cdot M$. The desired result then follows from the recollection that $\|g\|_p^p = M$. $\qquad\square$

We now show the correctness of [Algorithm 3](#).

**Lemma D.3.** *For any fixed time during a stream, let $f$ be the induced frequency vector and let $\widehat{F}$ be the output of [Algorithm 3](#). Then we have that with high probability,*

$$(1 - \varepsilon)\|f\|_p^p \leq \widehat{F} \leq (1 + \varepsilon)\|f\|_p^p.$$

*Proof.* Consider the first time $t$ in a block of $\ell$ updates and let $f$ be the frequency vector induced by the stream up to that point. We first observe that ROBUSTHH with threshold $\varepsilon\eta$ will return any coordinates $i \in [n]$ such that $f_i \geq \varepsilon^p\eta^p \cdot \|f\|_p^p$ up to $(1 + \varepsilon)$-approximation. For the remaining coordinates in the $k$-sparse vector returned by ROBUSTHH, any $k$ of them can contribute at most $\varepsilon^p \cdot \|f\|_p^p$. Therefore, we have by [Lemma D.1](#) that conditioned on the correctness of ROBUSTHH

and RESIDUALEST, we have $\widehat{G} + \widehat{H}$ is a $(1 + \mathcal{O}(\varepsilon))$-approximation to $\|f\|_p^p$. For the purposes of notation, let $h$ denote the residual vector of $f$ at time $t$, omitting the $k$ coordinates of $f$ largest in magnitude.

Now, consider some later time $t'$ in the same block of $\ell$ updates and let $v$ be the frequency vector induced by the updates in the block, i.e., the updates from $t$ to $t'$. Let $u$ be the residual vector omitting the $k$ coordinates of $f + v$ largest in magnitude. Since $\|v\|_1 \leq \ell$ for $\ell = \mathcal{O}\left(\varepsilon \cdot m^{c/p} k^{1-1/p}\right)$, then by Lemma 4.2, we have that $|\|h\|_p^p - \|u\|_p^p| \leq \frac{\varepsilon}{4} \cdot \|h\|_p^p$. Thus provided that $\widehat{H}$ is a $(1 + \mathcal{O}(\varepsilon))$-approximation to $\|h\|_p^p$, then it remains a $\left(1 + \frac{\varepsilon}{4}\right)$-approximation to $\|u\|_p^p$. Hence conditioned on the correctness again of ROBUSTHH at time $t'$, we have that $\widehat{H} + \widehat{H}$ remains a $(1 + \varepsilon)$-approximation to $\|f\|_p^p$ at time $t$.

As correctness of ROBUSTHH follows from Theorem 1.2, it remains to show correctness of RESIDUALEST on an adaptive stream. Because each block has size $\ell$, then the stream has at most $\frac{m}{\ell}$ such blocks. Hence by the adversarial robustness of differential privacy, i.e., Theorem 2.3, it suffices to run $\tilde{\mathcal{O}}\left(\frac{\sqrt{m}}{\ell}\right)$ copies of RESIDUALEST to guarantee correctness with high probability at all times. $\square$

Finally, we analyze the space complexity of our algorithm.

**Lemma D.4.** *For* $\log n = \Theta(\log m)$, *Algorithm 3 uses* $\tilde{\mathcal{O}}\left(\frac{1}{\varepsilon^{7.5}} \cdot m^c\right)$ *bits of space in total.*

*Proof.* We observe that Algorithm 3 uses a few main subroutines. Firstly, it runs SPARSERECOVER with sparsity $\mathcal{O}(m^c)$, which requires $m^c \cdot \text{polylog}(nm)$ bits of space, by Theorem 4.1. Next, it runs ROBUSTHH with threshold $\varepsilon\eta$, for $\eta = \frac{\varepsilon^2}{100m^\gamma}$. By Theorem 1.2, ROBUSTHH uses $\tilde{\mathcal{O}}\left(\frac{1}{(\varepsilon\eta)^{2.5}} m^{(2p-2)/(4p-3)}\right)$ bits of space. Note that for our choice of $\gamma = \frac{2c}{5} - \frac{(4p-4)}{(20p-15)}$, we have $\tilde{\mathcal{O}}\left(\frac{1}{(\varepsilon\eta)^{2.5}} m^{(2p-2)/(4p-3)}\right) = \tilde{\mathcal{O}}\left(\frac{1}{\varepsilon^{7.5}} \cdot m^c\right)$ bits of space. Finally, it runs $\tilde{\mathcal{O}}\left(\frac{\sqrt{m}}{\ell}\right)$ copies of LZEROEST and RESIDUALEST. By Theorem 3.4, each instance of RESIDUALEST uses $\tilde{\mathcal{O}}\left(\frac{1}{\varepsilon^6} \cdot \log^3(nm)\right)$ bits of space. By Theorem 2.4, each instance of LZEROEST uses $\mathcal{O}\left(\log^2(nm) \log \log m\right)$ bits of space. Since $\ell = \mathcal{O}\left(\varepsilon \cdot m^{c/p} k^{1-1/p}\right)$, then we have $\tilde{\mathcal{O}}\left(\frac{\sqrt{m}}{\ell}\right) = \tilde{\mathcal{O}}(m^c)$ and thus the total space usage by these subroutines is $\tilde{\mathcal{O}}\left(\frac{1}{\varepsilon^6} \cdot m^c\right)$ bits The desired claim then follows by noting that across all procedures, the space usage is $\tilde{\mathcal{O}}\left(\frac{1}{\varepsilon^{7.5}} \cdot m^c\right)$, due to our balancing choices of $\ell, \gamma$, and $c$. $\square$

**Lemma 4.3.** *For* $\log n = \Theta(\log m)$, *Algorithm 3 uses* $\tilde{\mathcal{O}}\left(\frac{1}{\varepsilon^{7.5}} \cdot m^c\right)$ *bits of space in total. Moreover, for any fixed time during a stream, let $f$ be the induced frequency vector and let $\widehat{F}$ be the output of Algorithm 3. Then we have that with high probability,* $(1-\varepsilon)\|f\|_p^p \leq \widehat{F} \leq (1+\varepsilon)\|f\|_p^p$.

*Proof.* Note that Lemma 4.3 follows from Lemma D.3 and Lemma D.4. $\square$

Putting things together, we get the full guarantees of our main result:

**Theorem 1.3.** *Let* $p \in [1,2]$ *and* $c = \frac{24p^2 - 23p + 4}{(4p-3)(12p+3)}$. *There exists a streaming algorithm that uses* $\mathcal{O}(m^c) \cdot \text{poly}\left(\frac{1}{\varepsilon}, \log(nm)\right)$ *bits of space and outputs a* $(1+\varepsilon)$-*approximation to the $L_p$ norm of the underlying vector at all times of an adversarial stream of length $m$.*

*Proof.* Observe that the correctness stems from Lemma D.3, while the space complexity follows form Lemma D.4. $\square$

## Broader Impact Statement

As adversarial robustness can have applications to many applications in machine learning, a potential broader impact of our work is the advancement of the theoretical foundations of trustworthy machine learning. There are many potential societal consequences of our work, none which we feel must be specifically highlighted here.

