# OpenReview forum: "Adversarially Robust Dense-Sparse Tradeoffs via Heavy-Hitters"
_NeurIPS.cc/2024/Conference — NeurIPS 2024 poster_

### Official Review · Reviewer_B1cC · 2024-07-11

**Soundness:** 3
**Presentation:** 3
**Contribution:** 3
**Rating:** 7
**Confidence:** 3

**Summary:**

The authors present an algorithm for adversarially robust $L_p$-estimation estimation in the turnstile streaming model, which improves on the space-complexity of existing algorithms in the regime $p \in (1, 2)$. They achieve this improvement via an alteration of the dense-sparse tradeoff technique of the existing state-of-the-art algorithm where they show it is sufficient to track the heavy-hitters while keeping track of an estimated residual. In addition to the improved Lp-estimation algorithm, the results include an adversarially-robust streaming algorithm for  the heavy-hitters problem along with an adversarially robust algorithm for estimating the residual of the frequency vector. The authors also include experiments providing support for their theoretical improvement guarantees.

**Strengths:**

- The authors do a great job of situating themselves in prior work, and explaining exactly how their results differ and improve.
- While I am not well-versed in this area, the results seem correct and were clearly explained and decomposed.

**Weaknesses:**

- The actual improvement over the previous work seems very minimal, i.e. a tiny reduction in the space complexity on a very small portion of possible choices of p. However, the relation to heavy-hitters seems like an interesting and non-trivial insight used to achieve this improvement, and the algorithms for heavy hitters and residual estimation seem to me to be of independent interest.

- The authors do a great job at explaining the intuition behind their techniques and how they fit with existing works. However, I found the actual preliminaries to be extremely confusing. The paper would greatly benefit from a clearer introduction on the exact setting and definitions of the various variables. I found the introduction of  https://arxiv.org/pdf/2003.14265 to be very helpful in clarifying my confusion.

**Questions:**

- What is the variable b that is referred to in Algorithm 3 (in the parameters for LZeroEst and ResidualEst)? Perhaps I am missing something but I don't see where it is defined.
- I was a bit confused on the purpose of the empirical evaluations. It seems that you demonstrate that empirically in one real-world instance, the flip number of the residual vector can be significanly smaller than the flip number of the entire datastream. However, if I am understanding correctly, your algorithmic guarantees depend on the worst-case flip-number of the residual vector, and are not adaptive to the actual flip number. How should I interpret these empirical results with respect to the performance of your proposed algorithm?

**Limitations:**

I think the authors have adequately addressed limitations and the assumptions made in their results.

---

> ### Author Rebuttal · Authors · 2024-08-06
>
> > The actual improvement over the previous work seems very minimal, i.e. a tiny reduction in the space complexity on a very small portion of possible choices of p. However, the relation to heavy-hitters seems like an interesting and non-trivial insight used to achieve this improvement, and the algorithms for heavy hitters and residual estimation seem to me to be of independent interest.
>
> Indeed, we do agree that in some settings, the quantitative improvement is not large. However, we emphasize that optimal bounds for adversarial insertion-deletion streams is a major open question and therefore, the main strength of our work is the conceptual message that the despite the lack of recent progress, previous results are not optimal. Thus we hope our work will inspire future research in this direction.
>
> > The authors do a great job at explaining the intuition behind their techniques and how they fit with existing works. However, I found the actual preliminaries to be extremely confusing. The paper would greatly benefit from a clearer introduction on the exact setting and definitions of the various variables. I found the introduction of https://arxiv.org/pdf/2003.14265 to be very helpful in clarifying my confusion.
>
> Thanks for the suggestion. We will expand the preliminaries to reiterate the model described in Section 1 (lines 50-67) with the discussion specifically centered around the $L_p$ heavy-hitter problem and/or the $L_p$ norm estimation problem.
>
> > What is the variable b that is referred to in Algorithm 3 (in the parameters for LZeroEst and ResidualEst)? Perhaps I am missing something but I don't see where it is defined.
>
> Similar to Algorithm 1, $b$ is the number of queries made to the oblivious algorithms. Since the stream has length $m$ and the stream is split into blocks of length $\ell$, then there are at most $\frac{m}{\ell}$ queries made to each of the algorithms, as in Algorithm 1. We will explicitly clarify the setting of $b=\frac{m}{\ell}$ in Algorithm 3.
>
> > I was a bit confused on the purpose of the empirical evaluations. It seems that you demonstrate that empirically in one real-world instance, the flip number of the residual vector can be significanly smaller than the flip number of the entire datastream. However, if I am understanding correctly, your algorithmic guarantees depend on the worst-case flip-number of the residual vector, and are not adaptive to the actual flip number. How should I interpret these empirical results with respect to the performance of your proposed algorithm?
>
> Yes, that's a fair point. However, we remark that the algorithmic guarantees of the previous results also depend on the worst-case flip-number of the entire vector, so in some sense we are still comparing apples to apples by comparing the empirical flip-number of the residual vector to the empirical flip-number of the entire vector. In particular, the algorithm must still commit to some budget for the flip-number before the algorithm is initialized and our empirical results show that the same budget can handle significantly more updates for our algorithm.

---

> > ### Comment · Reviewer_B1cC · 2024-08-07
> >
> > Thanks for your response! I will tentatively raise my score to a 7: Accept.

---

### Official Review · Reviewer_VAPi · 2024-07-12

**Soundness:** 4
**Presentation:** 4
**Contribution:** 3
**Rating:** 7
**Confidence:** 4

**Summary:**

In adversarially robust streaming, one wants to design streaming algorithms that work well even in the interactive setting, in which the stream is not fully fixed in advance, but is constructed element by element by an adversary who can see the current solution provided by the streaming algorithm. If the algorithm is randomized, each intermediate solution may reveal some information about the internal randomness of the car, which could allow the adversary make updates that break the estimates of the algorithm.

The paper concerns two fundamental problems in this setting: heavy hitters and moment estimation. In particular, the paper improves on the best known algorithm for moment estimation that considers two regimes, sparse and dense vectors. The paper shows that estimating $L_p$ moments for $p \in [1,2)$ can be done in smaller space than previously known.

The achievement follows by leveraging known deterministic algorithms for heavy hitters and using them for tracking significant changes to the frequency vector.

**Strengths:**

* Insightful contribution to a very active research area.

* Non-trivial improvement for some of the most popular streaming problems: heavy hitters and moments. The combination of different ideas to make this work is not easy.

**Weaknesses:**

The moments/norms for which the paper improves the space requirements are not the most important ones, which are I think are $p=0$ and $p=2$.

**Questions:**

"Unusual" values of moments can be used for computing entropy (see Harvey, Nelson, Onak 2008). Do you think this could be used here to provide further motivation for your results?

Your result heavily relies on Ganguly's heavy hitters result. I tried looking at this and related papers but I didn't have enough time to read them. Those are not the most frequently cited papers. Did you maybe verify their correctness?

Are your results in the general turnstile model (where frequencies can get negative) or the strict turnstile modeli (where deletions are allowed, but all frequencies are always non-negative)? In particular, this goes back to the results of Ganguly. Which of the models are they for? Because if they are only for strict turnstile, then the case of $p=1$ is trivial.

**Limitations:**

No direct negative social impact.

---

> ### Author Rebuttal · Authors · 2024-08-06
>
> > The moments/norms for which the paper improves the space requirements are not the most important ones, which are I think are $p=0$ and $p=2$.
>
> In general, a common goal is to find the heavy-hitters above a desired threshold that may be arbitrary. In this case, we want $|x_i|>\tau$ for some threshold $\tau$. We can then choose the values of $\varepsilon$ and $p$ accordingly, so that $\tau=\varepsilon\cdot\|x\|_p$ so that the $L_p$-heavy hitters correspond to the items whose frequency are above the threshold.
>
> > "Unusual" values of moments can be used for computing entropy (see Harvey, Nelson, Onak 2008). Do you think this could be used here to provide further motivation for your results?
>
> Yes, entropy estimation is certainly an additional application for our results, particularly since the previous streaming algorithm utilizes $F_p$ moment estimation for $p\in(1,2)$. Thanks for the pointer!
>
> > Your result heavily relies on Ganguly's heavy hitters result. I tried looking at this and related papers but I didn't have enough time to read them. Those are not the most frequently cited papers. Did you maybe verify their correctness?
>
> > Are your results in the general turnstile model (where frequencies can get negative) or the strict turnstile modeli (where deletions are allowed, but all frequencies are always non-negative)? In particular, this goes back to the results of Ganguly. Which of the models are they for? Because if they are only for strict turnstile, then the case of $p=1$ is trivial.
>
> Ganguly and Majumder's heavy-hitters algorithm uses Chinese remainder codes in their Lemma 9 statement and can be stated for the general turnstile model. Importantly, their Lemma 11 does not give an $\Omega(n)$ lower bound for the general turnstile model for the heavy-hitter problem because it only applies for heavy-hitters problem "with parameter $s$" with $s=\Omega(n)$, where the additive error for each estimate is $\frac{||x||_1}{s}$ and the universe has size $n$.
>
> Furthermore, their work is subsequently extended to general error correcting codes in [NNW12], which actually achieves a slightly better space bound in logarithmic dependencies. Specifically, [NNW12] constructs a deterministic matrix $A$ with Johnson-Lindenstrauss properties, so that maintaining $Ax$ for the underlying frequency vector $x$ is amenable even in the general turnstile model.
>
> [NNW12] Jelani Nelson, Huy L. Nguyên, David P. Woodruff: On Deterministic Sketching and Streaming for Sparse Recovery and Norm Estimation. APPROX-RANDOM 2012: 627-638

---

> > ### Comment · Reviewer_VAPi · 2024-08-07
> >
> > Thank your for the response and reassuring me about the correctness of papers by Ganguly et al. I'm happy to improve my score to 7 (Accept). I still stand behind my conviction that the cases of $p = 0$ or $p =2$ would be more interesting.
> >
> > As for the entropy application question, let me just warn you about a subtle point if you decide to mention it in your paper. For a fixed constant $p$, you usually do not care if there is a multiplicative factor that depends only on $p$ in the complexity of estimating $\ell_p$. In the entropy application, you are not considering, however, a fixed $p$, but you select $p = 1 + \delta$, where $\delta$ is some function of $\epsilon$ and perhaps parameters of the stream. Then the final complexity will have an additional multiplicative factor that depends on $\delta$ and you have to make sure it's not prohibitively large. This is usually not what people focus on for moment estimation, but for standard moment estimation algorithms, you get at a factor of at least $1/\delta$ I think. You would have to check the literature devoted to fractional moment estimation and see what impact (if any) this has on the complexity of deterministic heavy hitters you use.

---

### Official Review · Reviewer_pbML · 2024-07-12

**Soundness:** 3
**Presentation:** 3
**Contribution:** 3
**Rating:** 8
**Confidence:** 4

**Summary:**

This paper focuses on the L_p estimation (of the frequencies of items in a stream) in the adversarially robust streaming setting. The previous work by Ben-Eliezer, Eden and Onak achieved an $\tilde{O}(m^{p/2p+1})$ space, which is slightly better than the $P(\sqrt{m})$ space bound due to the flip number technique by considering a sparse-dense tradeoff. The authors of this work question whether the above bound is tight or if it can be improved.

The authors present an algorithm that beats the bound of $\tilde{O}(m^{p/2p+1})$. This is obtained by first building an adversarially robust streaming algorithm for L_p heavy hitters, utilizing deterministic turnstile heavy-hitter algorithms with better tradeoffs. This is then combined with another algorithm that estimates the frequency of the tail-vector, and has additive error and space independent of the size of the tail.

**Strengths:**

As the paper mentions, this is a conceptual breakthrough, showing that the previous bound of $\tilde{O}(m^{p/2p+1})$ is not tight. The paper explains all the ideas pretty well, and it was easy to read.

The novel insight here is that in the dense-sparse tradeoff of [BEO22], in order to change the p-th moment by at least a $1+\epsilon$ factor, most of the updates are to a small number of coordinates, which then naturally must have either already been heavy-hitters, or became so after the update. The authors are able to effectively handle the had case of [BEO22], resulting in a better improvement.

**Weaknesses:**

The only obvious weakness is that the result is a very slight improvement, sometimes in the third decimal place. While the insight is nice, the techniques are a linear combination of different algorithms for different cases.  While I did not find the techniques very exciting, I think this passes the bar for NeurIPS.

**Questions:**

Do you see any potential barriers to this approach? A discussion on what the limits are (is the bottleneck the heavy hitters algorithm or the tail-estimation algorithm?) would be greatly appreciated. This will also give the reader the assurance that this minor improvement is still significant as it pushes a new idea to a reasonable extent.

**Limitations:**

Apart from the limitation described in the theorem statement, the authors could discuss limitations of this approach, and what the current barrier is.

---

> ### Author Rebuttal · Authors · 2024-08-06
>
> > The only obvious weakness is that the result is a very slight improvement, sometimes in the third decimal place. While the insight is nice, the techniques are a linear combination of different algorithms for different cases. While I did not find the techniques very exciting, I think this passes the bar for NeurIPS.
>
> We do agree that in some settings, the quantitative improvement is not large. However, we emphasize that optimal bounds for adversarial insertion-deletion streams is a major open question and therefore, the main strength of our work is the conceptual message that the despite the lack of recent progress, previous results are not optimal. Thus we hope our work will inspire future research in this direction.
>
> > Do you see any potential barriers to this approach? A discussion on what the limits are (is the bottleneck the heavy hitters algorithm or the tail-estimation algorithm?) would be greatly appreciated. This will also give the reader the assurance that this minor improvement is still significant as it pushes a new idea to a reasonable extent.
>
> The relatively larger space used by the deterministic heavy-hitter algorithm compared to the space-optimal randomized heavy-hitter algorithms is a major bottleneck on further improving the bounds for this approach. One hope would be to improve the space usage of the deterministic heavy-hitter algorithm. However, it is known that the existing bounds are optimal, i.e., there are no deterministic heavy-hitter algorithms that use less space. Thus, the current analysis we use is tight for this approach and any further improvements likely require significantly new or different algorithmic designs.

---

> > ### Comment · Reviewer_pbML · 2024-08-10
> >
> > Thank you for the response.

---

### Official Review · Reviewer_KENe · 2024-07-13

**Soundness:** 4
**Presentation:** 2
**Contribution:** 3
**Rating:** 6
**Confidence:** 3

**Summary:**

The work develops new algorithms in the adversarially robust streaming model. In this model, the algorithm observes updates to some vector arrive in the form of a data stream and maintain estimates to some property of the vector as it changes. At each time-step i, the algorithm receives the update of the vector at this time-step. Specifically, it receives an update of the form $(i, \Delta_i)$ that means that the $i$-th coordinate of the vector changes by $\Delta_i$. Also, at every time-step i the algorithm outputs an estimate of some property of this vector.

 The work focuses on the adversarially robust setting, where the updates in subsequent time-steps can potentially depend on the estimates that the algorithm gave in previous time-steps. This is an important property of the streaming algorithm, because when decisions are made based on the estimates given by a streaming algorithm this will likely influence the data received by this algorithm in the future. At the same time, notably, many previously studied streaming algorithms are not adversarially robust.
Additionally, the paper focuses on the turnstile setting, i.e. the updates $\Delta_i$ to the vector can be both positive and negative.

The length of the data stream is denoted by $m$, and in the adversarially robust streaming setting the paper mainly studies the problems of $L_p$-heavy hitter estimation and the problem of estimating the $L_p$ norm of the vector.

The problem of adversarially robust  $L_p$-heavy-hitter estimation requires the streaming algorithm maintain a list of elements that contribute a lot to the $L_p$ norm of the vector. Formally, it requires the list to contain all elements $i$ for which $|x_i| \geq \epsilon ||x||_p$ and all elements $j$ in the list should satisfy $|x_j| \geq \epsilon/2 ||x||_p$.

Up to constantants and logarithmic factors in m and $\epsilon$, the $L_p$-heavy-hitter estimation algorithm requires $1/\epsilon^2.5 * m^\alpha$, where $\alpha=(2p-2)/(4p-3)$. This improves on the previous algorithm from the literature that achieves $\alpha=p/(2p+1)$. This constitutes an improvement for all $p$ in $[1,2)$, with the most pronounced improvement at $p=1$ where the space usage is improved from polynomial in $m$ to polylogarithmic.

The work also presents new algorithms for estimating the $L_p$-norm of the underlying vector (up to a multiplicative factor of $1+\epsilon$). Up to polylogarithmic factors and constants, the amount of space used by the algorithm is of the form $m^c poly(1/\epsilon)$. The exact dependence of $c$ on the value of $p$ is $c=(24p^2-23p+4)/((4p-3)(12p+3))$. As the paper notes, for $p=1.5$ we have $c=0.373$ whereas the best previous algorithm has $c=0.375$.

The work improves on the dense-sparse decomposition method of the previous work.

**Strengths:**

- The adversarially robust streaming framework is natural and well-motivated.

- For the L_1-heavy-hitter problem the work gives qualitative improvement on the previous algorithms by improving the space use from polynomial in the stream length to polylogarithmic.

**Weaknesses:**

- Other than the $L_1$-heavy hitter estimation, many of the improvements are extremely incremental, for instance the aforementioned improvement of the dependence on the length $m$ of the stream for $L_{1.5}$-norm estimation to $m^{0.373}$ from $m^{0.375}$.

- Other than the results for L_1-heavy-hitter estimation, my understanding is that there is no evidence of optimality for the results given in this work. For example, to the best of my understanding, it can conceivably be the case that the above-mentioned dependence of  $m^{0.373}$  could in the future be improved to polylog$(m)$. If this happens, the impact of this work could potentially be small.

**Questions:**

In subsection 1.1 what is the relationship between vectors $f$ and $x$? In the rest of the paper $f$ is used to denote the frequency vector of the stream, whereas the vector $x$ does not appear to be invoked anywhere again. In any case, it would be helpful if the vectors $x$ and $f$ are clearly defined in the beginning of subsection 1.1.

Subsection 1.1. would be easier to read if the paper used \paragraph to separate the parts of the subsection that address different problems.

**Limitations:**

It would be helpful if the authors added a dedicated paragraph in the paper that indicated what are the main limitations of the work according to the authors.

---

> ### Author Rebuttal · Authors · 2024-08-06
>
> > Other than the $L_1$-heavy hitter estimation, many of the improvements are extremely incremental, for instance the aforementioned improvement of the dependence on the length $m$ of the stream for $L_{1.5}$-norm estimation to $m^{0.373}$ from $m^{0.375}$.
>
> > Other than the results for L_1-heavy-hitter estimation, my understanding is that there is no evidence of optimality for the results given in this work. For example, to the best of my understanding, it can conceivably be the case that the above-mentioned dependence of $m^{0.373}$ could in the future be improved to polylog$(m)$. If this happens, the impact of this work could potentially be small.
>
> We do agree that in some settings, the quantitative improvement is not large. However, we emphasize that optimal bounds for adversarial insertion-deletion streams is a major open question and therefore, the main strength of our work is the conceptual message that the despite the lack of recent progress, previous results are not optimal. Thus we hope our work will inspire future research in this direction.
>
> > In subsection 1.1 what is the relationship between vectors $f$ and $x$? In the rest of the paper $f$ is used to denote the frequency vector of the stream, whereas the vector $x$ does not appear to be invoked anywhere again. In any case, it would be helpful if the vectors $x$ and $f$ are clearly defined in the beginning of subsection 1.1.
>
> Due to the problems of heavy-hitters and moment estimation, we use both $f$ and $x$ to denote the same underlying vector. We will unify this notation with the same symbol in the updated version.
>
> > Subsection 1.1. would be easier to read if the paper used \paragraph to separate the parts of the subsection that address different problems.
>
> Thanks for the suggestion, we will add paragraph notation to demarcate the discussion on heavy-hitters and norm estimation in the updated version.

---

> > ### Comment · Reviewer_KENe · 2024-08-07
> >
> > Thank you for your response. Tentatively, I keep my rating at 6: Weak Accept

---

### Author Rebuttal · Authors · 2024-08-06

We thank the reviewers for their thoughtful comments and valuable insight. We also appreciate the positive feedback, such as:
- The adversarially robust streaming framework is natural and well-motivated. (Reviewer KENe)
- For the L_1-heavy-hitter problem the work gives qualitative improvement on the previous algorithms by improving the space use from polynomial in the stream length to polylogarithmic. (Reviewer KENe)
- As the paper mentions, this is a conceptual breakthrough, showing that the previous bound of $\tilde{O}(m^{p/2p+1})$ is not tight. (Reviewer pbML)
- The paper explains all the ideas pretty well, and it was easy to read. (Reviewer pbML)
- The authors are able to effectively handle the had case of [BEO22], resulting in a better improvement. (Reviewer pbML)
- Insightful contribution to a very active research area. (Reviewer VAPi)
- Non-trivial improvement for some of the most popular streaming problems: heavy hitters and moments. The combination of different ideas to make this work is not easy. (Reviewer VAPi)
- The authors do a great job of situating themselves in prior work, and explaining exactly how their results differ and improve. (Reviewer B1cC)
- The results seem correct and were clearly explained and decomposed. (Reviewer B1cC)

We provide specific responses to the initial comments of each reviewer below. We hope our answers addresses all points raised by the reviewers and we will be most happy to answer any remaining or additional questions during the discussion phase!

---

### Decision · Program_Chairs · 2024-09-25

**Decision:**

Accept (poster)

**Comment:**

This paper studies the problem of $L_p$ heavy-hitters estimation in the robust streaming model. The results of the paper improve polynomially the memory requirements for this task, and the gains are more pronounced for $p=1$, where the memory bound becomes poly-logarithmic, as opposed to previous works where the corresponding bound is $O(m^{1/3})$. These results are based on novel innovations on previous works which introduced a dense-sparse tradeoff technique and techniques from differential-privacy. Reviewers were consistently positive about the merits of the work, while there were some concerns about the broader implications (or lack thereof) of these results, as well as the lack of matching lower bounds.